# Scalable Simulation-free Entropic Semi-Unbalanced Optimal Transport

## Abstract

The Optimal Transport (OT) problem investigates a transport map that connects two distributions while minimizing a given cost function. Finding such a transport map has diverse applications in machine learning, such as generative modeling and image-to-image translation. In this paper, we introduce a scalable and simulation-free approach for solving the Entropic Semi-Unbalanced Optimal Transport (ESUOT) problem. We derive the dynamical form of this ESUOT problem, which is a generalization of the Schrödinger bridges (SB) problem. Based on this, we derive dual formulation and optimality conditions of the ESUOT problem from the stochastic optimal control interpretation. By leveraging these properties, we propose a simulation-free algorithm to solve ESUOT, called Simulation-free ESUOT (SF-ESUOT). While existing SB models require expensive simulation costs during training and evaluation, our model achieves simulation-free training and one-step generation by utilizing the reciprocal property. Our model demonstrates significantly improved scalability in generative modeling and image-to-image translation tasks compared to previous SB methods.

## 1 Introduction

The distribution transport problem investigates finding a transport map that bridges one distribution to another. The distribution transport problem has various applications in machine learning, such as generative modeling (Rout et al., 2022; Choi et al., 2023a; 2024; Shi et al., 2024; Dao et al., 2023; Lübeck et al., 2022), image-to-image translation (Kim et al., 2024a), and biology (Bunne et al., 2023b; 2022). Optimal Transport (OT) (Peyré et al., 2019; Villani et al., 2009) explores the most cost-efficient transport map among them. For discrete measures, the OT map can be computed exactly through convex optimization, but it is computationally expensive. In contrast, the Entropic Optimal Transport (EOT) problem presents strict convexity and can be computed more efficiently through the Sinkhorn algorithm (Sinkhorn, 1967; Cuturi, 2013). For continuous measures, several machine learning approaches have been proposed for learning the EOT problem (Chen et al., 2021a; Shi et al., 2024; Gushchin et al., 2024). These approaches typically utilize a dynamic version of EOT, known as the Schrödinger Bridge problem (Léonard, 2013; Shi et al., 2024).

The Schrödinger Bridge (SB) is a finite-time diffusion process that bridges two given distributions while minimizing the KL divergence to a reference process (Chen et al., 2021b; Léonard, 2013). Various works for solving this SB problem have been proposed (Stromme, 2023; Finlay et al., 2020; Bortoli et al., 2021; Chen et al., 2021a; Shi et al., 2024). However, these methods tend to exhibit scalability challenges when the source and target distributions are in high-dimensional spaces and the distance between them is large. Consequently, these approaches tend to be limited to low-dimensional datasets (Stromme, 2023; Finlay et al., 2020) or rely on a pretraining process for generative modeling task (Chen et al., 2021a; Shi et al., 2024). Moreover, existing approaches require simulation of the diffusion process, leading to excessive training and evaluation costs (Shi et al., 2024; Chen et al., 2021a; Gushchin et al., 2024).

In this paper, we propose an algorithm for solving the Entropic Semi-Unbalanced Optimal Transport (ESUOT) problem, called *Simulation-free ESUOT (SF-ESUOT)*. The ESUOT problem generalizes the EOT problem by relaxing the precise matching of the target distribution into soft matching through $f$-divergence minimization. Depending on the chosen divergence measure, the ESUOT problem can encompass the EOT problem. To solve this, our model is based on the stochastic optimal control

interpretation of the dual formulation of ESUOT. Compared to previous works, our model introduces a novel parametrization consisting of the static generator for path measure $\rho$ and the time-dependent value function $V$. This parametrization enables simulation-free training and one-step generation through the reciprocal property (Sec. 4). Our model achieves significantly improved scalability compared to existing SB models. Specifically, our model achieves a FID score of 3.02 with NFE 1 on CIFAR-10 without pretraining, which is comparable to the state-of-the-art results of the SB model with pretraining. For instance, IPF (Chen et al., 2021a) achieves a FID score of 3.01 with NFE 200 and IMF (Shi et al., 2024) achieves 4.51 with NFE 100. Furthermore, our model outperforms several OT models on image-to-image translation benchmarks. Our contributions can be summarized as follows:

- We derive the dynamical form of the Entropic Semi-Unbalanced Optimal Transport (ESUOT) problem, which is a generalization of the Schrödinger Bridge Problem.
- We derive the dual formulation of the ESUOT problem and its optimality conditions through the stochastic optimal control interpretation (Sec. 3).
- We propose an efficient method for solving ESUOT problem based on these interpretations. Our model offers simulation-free training by utilizing the reciprocal property (Sec. 4).
- Our model greatly improves the scalability of E(SU)OT models. To the best of our knowledge, our model is the first EOT model that presents competitive results without pretraining (Sec. 5).

**Notations and Assumptions** Let $\mathcal{X} = \mathbb{R}^d$ where $d$ is a data dimension. Let $\mathcal{M}_2(\mathcal{X})$ be a set of positive Borel measures with finite second moment. Moreover, let $\mathcal{P}_2(\mathcal{X})$ be a set of probability densities in $\mathcal{M}_2(\mathcal{X})$. Let $\Phi_2(\mathcal{X})$ be a set of continuous functions $\varphi : \mathcal{X} \to \mathbb{R}$ such that $|\varphi(x)| \leq a + b\|x\|^2$ for all $x \in \mathcal{X}$, for some $a, b > 0$. Let $\Phi_{2,b}$ be a set of bounded-below functions in $\Phi_2$. Throughout the paper, let $\mu, \nu \in \mathcal{P}_2(\mathcal{X})$ be the absolutely continuous source and target distributions, respectively. In generative modeling tasks, $\mu$ and $\nu$ correspond to *tractable noise* and *data distributions*. $\Pi(\mu, \nu)$ denotes the set of joint probability distributions on $\mathcal{P}_2(\mathcal{X} \times \mathcal{X})$ whose marginals are $\mu$ and $\nu$. $W_t$ represents the standard Wiener process on $\mathcal{X}$.

## 2 Background

In this section, we provide a brief overview of key concepts in the OT theory and the Schrödinger Bridge (SB) problem. For detailed discussion on related works, please refer to Appendix B.

### 2.1 Optimal Transport Problems

**Kantorovich's Optimal Transport (OT)** Kantorovich's Optimal Transport (OT) (Kantorovich, 1948) problem addresses the problem of searching for the most cost-effective way to transform the source distribution $\mu$ to the target distribution $\nu$. Formally, it can be expressed as the following minimization problem:

$$\inf_{\pi \in \Pi(\mu,\nu)} \int \frac{1}{2}\|x - y\|_2^2 d\pi(x,y) \tag{1}$$

Here, we consider the quadratic cost $\frac{1}{2}\|x - y\|_2^2$. For an absolutely continuous $\mu \in \mathcal{P}_2(\mathcal{X})$, the optimal transport plan $\pi^\star$ exists. Moreover, the optimal transport is deterministic. In other words, there exists a deterministic OT map $T^\star : \mathcal{X} \to \mathcal{X}$ such that $(Id \times T^\star)_{\#}\mu = \pi^\star$, i.e. $\pi^\star(\cdot|x) = \delta_{T^\star(x)}$ is a delta measure.

**Entropic Optimal Transport (EOT)** The Entropic Optimal Transport (EOT) problem is an entropy-regularized version of the OT problem, which introduces an entropy term for the coupling $\pi$ to the standard OT problem. Formally, EOT can be written as the following minimization problem:

$$\inf_{\pi \in \Pi(\mu,\nu)} \int_{\mathcal{X} \times \mathcal{X}} \frac{1}{2}\|x - y\|^2 d\pi(x,y) - \sigma^2 H(\pi), \tag{2}$$

where $H(\pi)$ denotes the entropy of $\pi$ and $\sigma > 0$. The optimal transport plan $\pi^\star$ of Eq. 2 is unique due to the strict convexity of $H(\pi)$. Moreover, when $\sigma > 0$, the optimal transport map is stochastic, i.e. $\pi^\star(\cdot|x)$ is a conditional probability distribution.

## 2.2 Schrödinger Bridge Problem and its Properties

In this section, we discuss the properties of the Schrödinger Bridge (SB) problem (Chen et al., 2021b; Léonard, 2013). First, we describe the reciprocal property of SB (Léonard et al., 2014). Then, we introduce the equivalence between the SB problem and EOT (Chen et al., 2021b; Léonard, 2013). Finally, we present the dual formulation of EOT (Gushchin et al., 2024). These properties will be extended to the Entropic Unbalanced Optimal Transport (EUOT) in Sec. 3.

**Schrödinger Bridge (SB) with Wiener prior** Let $\Omega := [0, 1] \times \mathcal{X}$, where $t \in [0, 1]$ represents the time variable. Throughout this paper, we denote the probability density induced by the following stochastic process $\{X_t^u\}$ as $\mathbb{P}^u \in \mathcal{P}_2(\Omega)$. In other words, $\mathbb{P}_t^u$ represents the distribution of the stochastic process $\{X_t^u\}$:

$$dX_t^u = u_t(X_t^u)dt + \sigma dW_t, \quad X_0^u \sim \mu. \tag{3}$$

with drift $u_t : \Omega \to \mathcal{X}$, diffusion term $\sigma > 0$, and initial distribution $\mu$. Moreover, let $\mathbb{Q}$ be the Wiener process with a diffusion term of $\sigma$. Then, the SB problem aims to find the probability density $\mathbb{P}^u$ that is most close to the Wiener process $\mathbb{Q}$. Formally, the SB problem is defined as follows:

$$\inf_u D_{\mathrm{KL}}(\mathbb{P}^u|\mathbb{Q}) \quad \text{s.t.} \quad \mathbb{P}_0^u = \mu, \ \mathbb{P}_1^u = \nu \quad \text{where} \quad d\mathbb{Q}_t = \sigma dW_t, \ \mathbb{Q}_0 \sim \mu. \tag{4}$$

Then, we can express this SB problem in terms of the drift $u$ and the path measure $\{\rho_t\}_{t \in [0,1]}$ of the stochastic process $X_t^u$. By Girsanov's theorem (Särkkä & Solin, 2019),

$$D_{\mathrm{KL}}(\mathbb{P}^u|\mathbb{Q}) = \frac{1}{\sigma^2} \mathbb{E}\left[\int_0^1 \frac{1}{2}\|u_t(X_t^u)\|^2 dt\right]. \tag{5}$$

Moreover, by the Fokker-Planck equation (Risken & Risken, 1996), $\{\rho_t\}_{t \in [0,1]}$ satisfies the following equation:

$$\partial_t \rho_t + \nabla \cdot (u_t \rho_t) - \frac{\sigma^2}{2} \Delta \rho_t = 0, \quad \rho_0 = \mu. \tag{6}$$

Therefore, by combining Eq. 5 and 6, the SB problem (Eq. 4) can be reformulated as follows:

$$\inf_u \left[\frac{1}{\sigma^2} \int_0^1 \int_{\mathcal{X}} \frac{1}{2}\|u_t(x)\|^2 d\rho_t(x)dt\right], \quad \text{s.t.} \ \partial_t \rho_t + \nabla \cdot (u_t \rho_t) - \frac{\sigma^2}{2}\Delta \rho_t = 0, \ \rho_0 = \mu, \rho_1 = \nu. \tag{7}$$

**Reciprocal Property of SB** In this paragraph, we provide an intuition of the Reciprocal Property (Léonard et al., 2014; Chen et al., 2021b; Shi et al., 2024) of SB. To begin with, the minimization objective of the SB problem in Eq. 4 can be decomposed into the KL divergence of the joint distribution between $t = 0$ and $t = 1$, and the conditional KL divergence. Formally, the decomposition is written as follows:

$$D_{\mathrm{KL}}(\mathbb{P}^u|\mathbb{Q}) = D_{\mathrm{KL}}(\mathbb{P}_{0,1}^u|\mathbb{Q}_{0,1}) + \int_{\mathcal{X} \times \mathcal{X}} \int_0^1 D_{\mathrm{KL}}(\mathbb{P}_t^u(\cdot|x,y)|\mathbb{Q}_t(\cdot|x,y)) \, dt \, d\mathbb{P}_{0,1}^u(x,y), \tag{8}$$

where $\mathbb{P}_{0,1}^u \in \Pi(\mu, \nu)$ denotes the joint distribution on $t = 0$ and $t = 1$ induced by $\mathbb{P}^u$. Let $\mathbb{P}^\star$ be the optimal solution for the LHS of Eq. 8. Surprisingly, Léonard et al. (2014) discovered that the last term on the RHS of Eq. 8 is zero for $\mathbb{P}^\star$, i.e., $\mathbb{P}_t^\star(\cdot|x,y) = \mathbb{Q}_t(\cdot|x,y)$ for $(x,y) \sim \mathbb{P}_{0,1}^\star$ almost surely. **This property allows us to characterize the bridge measure between $X_0^u = x$ and $X_1^u = y$.**

$$\mathbb{P}_t^\star(\cdot|x,y) = \mathbb{Q}_t(\cdot|x,y) = \mathcal{N}(\cdot|(1-t)x + ty, \sigma^2 t(1-t)I). \tag{9}$$

We refer to Eq. 9 as the *reciprocal property* of SB. This property will be utilized in our neural network parametrization in Sec. 4.

**Equivalence between SB and EOT** The reciprocal property establishes equivalence between SB and EOT (For rigorous explanation, see (Léonard, 2013; Chen et al., 2021b)). By definition, the KL divergence between two static transport plans is given as follows:

$$\sigma^2 D_{\mathrm{KL}}(\pi|\mathbb{Q}_{0,1}) = \int_{\mathcal{X} \times \mathcal{X}} \frac{1}{2}\|x - y\|^2 d\pi(x,y) - \sigma^2 H(\pi). \tag{10}$$

Here, the reciprocal property implies that $D_{\mathrm{KL}}(\mathbb{P}^\star|\mathbb{Q}) = D_{\mathrm{KL}}(\mathbb{P}_{0,1}^\star|\mathbb{Q}_{0,1})$ in Eq. 8. Combining this with Eq. 10, we can show that the dynamical SB problem (Eq. 4) is equivalent to the static EOT problem (Eq. 2):

$$\sigma^2 D_{\mathrm{KL}}(\mathbb{P}^\star|\mathbb{Q}) = \inf_{\pi \in \Pi(\mu,\nu)} \left[\int \frac{1}{2}\|x - y\|^2 d\pi(x,y) - \sigma^2 H(\pi)\right] \quad \text{where} \quad \pi^\star = \mathbb{P}_{0,1}^\star. \tag{11}$$

**Dual Form of EOT** Additionally, we introduce a dual form of the EOT problem presented in Gushchin et al. (2024). As described in Eq. 10, the entropy-regularized minimization objective in EOT can be interpreted as minimizing $D_{\text{KL}}(\pi|\mathbb{Q}_{0,1})$ with respect to the joint distribution $\pi \in \Pi(\mu,\nu)$. Inspired by the weak OT theory (Backhoff-Veraguas et al., 2019), Gushchin et al. (2024) derived the following dual form:

$$\inf_{\pi \in \Pi(\mu,\nu)} D_{\text{KL}}(\pi|\mathbb{Q}_{0,1}) = \sup_{V \in \Phi_{2,b}} \left[ \int_{\mathcal{X}} V^C(x)d\mu(x) - \int_{\mathcal{X}} V(y)d\nu(y) \right], \tag{12}$$

where $V^C(x) := \inf_{\rho_1 \in \mathcal{P}_2(\mathcal{X})} \left[ D_{\text{KL}}\left(\rho_1 \mid \mathbb{Q}_{1|0}(\cdot|x)\right) + \int_{\mathcal{X}} V(y)\nu(y) \right]$. By applying the Girsanov theorem (Eq. 5) to the KL divergence in $V^C$, Gushchin et al. (2024) obtain the following problem:

$$\sup_{V \in \Phi_{2,b}} \inf_{u} \left[ \mathbb{E}\left[ \int_0^1 \frac{1}{2}\|u_t(X_t^u)\|^2 dt + V(X_1^u) \right] - \int_{\mathcal{X}} V(y)d\nu(y) \right], \text{ s.t. } X_0^u \sim \mu, \; X_1^u \sim \nu. \tag{13}$$

## 3 DYNAMCIAL AND DUAL FORM OF ENTROPIC UNBALANCED OPTIMAL TRANSPORT

In this section, **we derive various formulations of the *Entropic Semi-Unbalanced Optimal Transport (ESUOT)*.** First, we introduce the ESUOT problem and derive its dynamical formulation (Theorem 3.2). The dynamical formulation of ESUOT encompasses the Schrödinger Bridge (SB) problem, which is a dynamical form of EOT (Sec. 2.2). Next, we derive the dynamical dual form of ESUOT by leveraging the SB theory (Theorem 3.4). The dynamical dual will play a crucial role in deriving our learning objective and we will utilize the reciprocal property from the dynamical for our simulation-free parametrization in Sec 4.2. For detailed proof of the theorems, refer to Appendix A.1.

**Entropic Semi-Unbalanced Optimal Transport (ESUOT)** In this paper, we consider the ESUOT problem with a fixed source measure constraint, i.e., $\pi_0 = \mu$ (Eq. 14). While EOT assumes precise matching of two measures, i.e., $\pi_0 = \mu, \pi_1 = \nu$ (Eq. 2), ESUOT relaxes the target marginal constraint using the $f$-divergence $D_\Psi$. This unbalanced variant of the optimal transport problem offers outlier robustness (Balaji et al., 2020; Choi et al., 2023a) and the ability to handle class imbalance in datasets (Eyring et al., 2024). Formally, the ESUOT problem is defined as follows:

$$\inf_{\pi_0 = \mu, \pi \in \mathcal{P}_2(\mathcal{X} \times \mathcal{X})} \left[ \int_{\mathcal{X} \times \mathcal{X}} \frac{1}{2}\|x-y\|^2 d\pi(x,y) - \sigma^2 H(\pi) + \alpha D_\Psi(\pi_1|\nu) \right], \tag{14}$$

where $\alpha > 0$ denotes the divergence penalization intensity and $\Psi : [0,\infty) \to [0,\infty]$ is assumed to be a convex, lower semi-continuous, non-negative function, and $\Psi(1) = 0$. We call $\Psi$ an *entropy function* of $D_\Psi$. Furthermore, due to the convexity of $\Psi$, the solution $\pi^\star$ of Eq. 14 is unique.

**Remark 3.1** (**ESUOT is a Generalization of EOT**). Suppose $\Psi$ is a convex indicator $\iota$ at $\{1\}$, i.e. $\iota(x) = 0$ if $x = 1$ and $\iota(x) = \infty$ otherwise. Then, the corresponding $f$-divergence $D_\iota(\rho_1|\rho_2)$ equals 0 if $\rho_1 = \rho_2$ almost surely and $\infty$ otherwise. To obtain a finite objective in Eq. 14, it must hold $\pi_1 = \nu$ almost surely. Therefore, when $\Psi = \iota$, ESUOT becomes EOT.

**Dynamical Formulation of ESUOT** The following theorem proves the **dynamical formulation of the ESUOT problem**. Note that, compared to the dynamical formulation for EOT (SB) (Eq. 7), the minimization objective (Eq. 15) contains an additional divergence penalization term. Moreover, the theorem states that the optimal path measure $\mathbb{P}^\star$ for Eq. 15 satisfies the reciprocal property (Eq. 16), as in the SB problem. **This reciprocal property will be leveraged for the simulation-free parametrization** in Sec 4.2.

**Theorem 3.2.** *The ESUOT problem is equivalent to the following dynamical transport problem:*

$$\inf_{u} \left[ \int_0^1 \int_{\mathcal{X}} \frac{1}{2}\|u_t(x)\|^2 d\rho_t(x)dt + \alpha D_\Psi(\rho_1|\nu) \right], \tag{15}$$

*where $\partial_t \rho_t + \nabla \cdot (u_t \rho_t) - \frac{\sigma^2}{2}\Delta \rho_t = 0$ and $\rho_0 = \mu$. Moreover, the optimal solution $\mathbb{P}^\star$ satisfies the reciprocal property, i.e.,*

$$\mathbb{P}_t^\star(\cdot|x,y) = \mathcal{N}(\cdot|(1-t)x + ty, \sigma^2 t(1-t)I), \quad (x,y) \sim \mathbb{P}_{0,1}^\star\text{-almost surely.} \tag{16}$$

**Remark 3.3** (**Dynamic form of ESUOT is a Generalization of SB**). As discussed in Remark 3.1, suppose the entropy function $\Psi$ is the convex indicator $\iota$. By the same argument, $\rho_1 = \nu$ almost surely. Therefore, the dynamical ESUOT (Eq. 15) becomes equivalent to the SB (Eq. 7) when $\Psi = \iota$.

---

**Algorithm 1** Simulation-free ESUOT Algorithm

---

**Require:** The source distribution $\mu$ and the target distribution $\nu$. Convex conjugate of entropy function $\Psi^*$. Generator network $T_\theta$ and the value network $v_\phi : [0,1] \times \mathcal{X} \to \mathbb{R}$. Time (sampling) distribution $\mathcal{T}$. Total iteration number $K$.

1: **for** $k = 0, 1, 2, \ldots, K$ **do**
2:      Sample a batch $x \sim \mu, y \sim \nu, t \sim \mathcal{T}, z, \eta_1, \eta_2 \sim \mathcal{N}(\mathbf{0}, \mathbf{I})$.
3:      $\hat{y} \leftarrow T_\theta(x, z)$
4:      $x_t \leftarrow (1-t)x + t\hat{y} + \sigma\sqrt{t(1-t)}\eta_1$.
5:      $u_t \leftarrow \frac{\hat{y} - x_t}{1-t}, \sigma_t \leftarrow \sigma\sqrt{\frac{(1-t-\Delta t)\Delta t}{1-t}}$.
6:      $x_{t+\Delta t} \leftarrow x_t + u_t\Delta t + \sigma_t\eta_2$.
7:      $R \leftarrow \frac{v_\phi(t+\Delta t, x_{t+\Delta t}) - v_\phi(t, x_t)}{\Delta t} - \frac{\alpha}{2}\|\nabla v_\phi(t, x_t)\|^2 + \frac{\sigma^2}{2}\Delta v_\phi(t, x)$.
8:      Update $v_\phi$ by the following loss: $\lambda_D\|R\|^p - \frac{\alpha}{2}\|\nabla v_\phi(t, x_t)\|^2 - v_\phi(1, \hat{y}) + \Psi^*(v_\phi(1, y))$.
9:      Update $T_\theta$ by the following loss: $\lambda_G R$.
10: **end for**

---

**Dynamical Dual form of ESUOT** By applying the Fenchel-Rockafellar theorem (Singer, 1979) and leveraging the dual form in Gushchin et al. (2024), we derive the following dynamical dual formulation of ESUOT:

**Proposition 3.4** (**Dual formulation of ESUOT**). *The static ESUOT problem (Eq. 14) is equivalent to the following problem:*

$$\sup_{V \in \Phi_{2,b}} \inf_u \left[ \mathbb{E}\left[ \int_0^1 \frac{1}{2}\|u_t(X_t^u)\|^2 dt + V(X_1^u) \right] - \int_{\mathcal{X}} \alpha\Psi^*\left( \frac{V(y)}{\alpha} \right) d\nu(y) \right]. \quad (17)$$

*where $dX_t^u = u_t(X_t^u)dt + \sigma dW_t$ and $X_0^u \sim \mu$.*

Note that when the entropy function $\Psi$ is a convex indicator $\iota$, the convex conjugate $\Psi^*(x) = x$. In this case, the dynamical dual form of ESUOT (Eq. 17) reduces to Eq. 13. Therefore, Prop. 3.4 is an extension of the dual form of EOT. This dual formulation and its interpretation via stochastic optimal control will be utilized to derive our learning objective in Sec 4.2.

# 4 METHOD

In this section, we propose our scalable and simulation-free algorithm for solving the ESUOT problem. In Sec 4.1, we reinterpret our dual form (Eq. 17) through various formulations. Specifically, starting with the stochastic optimal control (SOC) interpretation, we interpret Eq. 17 as a bilevel optimization problem, involving the time-dependent value function $V$ and the path measure of $X_t^u$, $\rho : [0,1] \times \mathcal{X} \to \mathbb{R}$. In Sec. 4.2, we derive our **Simulation-free ESUOT** algorithm by integrating the results from Sec. 4.1 and the reciprocal property in Theorem 3.2.

## 4.1 STOCHASTIC OPTIMAL CONTROL INTERPRETATION OF DUAL FORM OF ESUOT

**SOC Interpretation of Inner-loop Problem** First, we investigate the inner optimization with respect to $u$ in the dual form of ESUOT (Eq. 17). By applying the SOC interpretation to this inner problem, we extend the value function $V : \mathcal{Y} \to \mathbb{R}$ to be time-dependent. The inner optimization of the dual form of ESUOT is given as follows:

$$\inf_u \mathbb{E}\left[ \int_0^1 \frac{1}{2}\|u_t(X_t^u)\|^2 dt + V(X_1^u) \right]. \quad (18)$$

This optimization problem can be regarded as a stochastic optimal control (SOC) problem (Fleming & Rishel, 2012) by regarding $X_t^u$ as the controlled SDE and $V(\cdot)$ as the terminal cost. Now, let $\tilde{V} : [0,1] \times \mathcal{X} \to \mathbb{R}$ be the **value function** of this problem, i.e.

$$\tilde{V}(t, x) = \inf_u \mathbb{E}\left[ \int_t^1 \frac{1}{2}\|u_t\|^2 dt + V(X_1^u) \mid X_t^u = x \right]. \quad (19)$$

Then, the inner minimization problem (Eq. 18) can be reformulated by the Hamilton-Jacobi-Bellman (HJB) equation for this time-dependent value function $\tilde{V}$ (Nüsken & Richter, 2021; Yong & Zhou, 2012):

$$\partial_t \tilde{V}_t - \frac{1}{2}\|\nabla \tilde{V}_t\|^2 + \frac{\sigma^2}{2}\Delta \tilde{V}_t = 0, \quad \tilde{V}_1 = V \quad \text{and} \quad u^\star = -\nabla \tilde{V}. \tag{20}$$

where $u^\star$ denotes the optimal control in Eq. 18. Since $\tilde{V}_1 := \tilde{V}(1, \cdot) = V$, we can say $\tilde{V}$ is an time-dependent extension of $V$. For simplicity, we will denote the time-dependent value function $\tilde{V}$ using the same notation as the original value function, specifically $V : [0, 1] \times \mathcal{X} \to \mathbb{R}$.

**Reinterpretation of Dual Form** We begin by interpreting the dual form of ESUOT (Eq. 17) as a bilevel optimization problem. **Our goal is to represent this optimization problem concerning the value function $V$ and the path measure $\rho_t$.** By splitting Eq. 17 into inner-minimization and outer-maximization and expressing $X_t^u$ using $\rho_t = \text{Law}(X_t^u)$ by the Fokker-Plank equation, we arrive at the following optimization problem:

$$\sup_{V \in \Phi_{2,b}} \int_0^1 \int_{\mathcal{X}} \frac{1}{2}\|u_t^\star\|^2 \rho_t^\star dx dt + \mathbb{E}_{\hat{y} \sim \rho_1^\star}[V_1(\hat{y})] - \int_{\mathcal{X}} \alpha \Psi^* \left( \frac{V_1(y)}{\alpha} \right) d\nu(y). \tag{21}$$

$$\text{s.t. } (u^\star, \rho^\star) = \arg \inf_{(u,\rho)} \mathbb{E}_{(t,x) \sim \rho} \left[ \frac{1}{2}\|u_t(x)\|^2 \right] + \mathbb{E}_{\hat{y} \sim \rho_1}[V_1(\hat{y})]. \tag{22}$$

where $\partial_t \rho + \nabla \cdot (u\rho) - (\sigma^2/2)\Delta \rho = 0$, $\rho_0 = \mu$. Finally, by incorporating the HJB optimality condition (Eq. 20) into (Eq. 22 and using $u^\star = -\nabla V$, we can represent the entire bilevel optimization problem above with respect to $V$ and the optimal $\rho_t^\star$ as follows:

$$\sup_{V \in \Phi_{2,b}} \int_0^1 \int_{\mathcal{X}} \frac{1}{2}\|\nabla V_t\|^2 \rho_t^\star dx dt + \mathbb{E}_{\hat{y} \sim \rho_1^\star}[V_1(\hat{y})] - \int_{\mathcal{X}} \alpha \Psi^* \left( \frac{V_1(y)}{\alpha} \right) d\nu(y).$$

$$\text{s.t. (HJB)} \quad \partial_t V_t - \frac{1}{2}\|\nabla V_t\|^2 + \frac{\sigma^2}{2}\Delta V_t = 0 \quad \rho^\star\text{-a.s.}, \tag{23}$$

$$\text{(Fokker-Plank)} \quad \partial_t \rho_t^\star + \nabla \cdot (-\nabla V_t \rho_t^\star) - \frac{\sigma^2}{2}\Delta \rho_t^\star = 0, \quad \rho_0^\star = \mu..$$

Note that we need to derive another optimization problem for the optimal $\rho_t^\star$ with respect to $V$. This can be achieved by considering the dual form of Eq. 22 as follows:

$$\inf_\rho \mathcal{L}_\rho \quad \text{where} \quad \mathcal{L}_\rho = \left[ \int_0^1 \int_{\mathcal{X}} \left( \partial_t V_t - \frac{1}{2}\|\nabla V_t\|^2 + \frac{\sigma^2}{2}\Delta V_t \right) \rho_t dx dt \right]. \tag{24}$$

For additional details on the derivation Eq. 24, please refer to Appendix A.2.

## 4.2 SIMULATION-FREE EUOT

In this section, we propose our algorithm for solving the ESUOT problem (Eq. 14), called **Simulation-free ESUOT (SF-ESUOT)**. Specifically, we optimize the dynamic value function $V$ and the path measure $\rho_t$. Our model is derived from two optimization problems (Eq. 23 and 24) for $V$ and $\rho_t$ presented in Sec 4.1. Additionally, we introduce a simulation-free parametrization of $\rho_t$ using the reciprocal property (Theorem 3.2), leading to Algorithm 1.

**Optimization of Value $V$** We present our loss function for the value function $V$, which is derived from Eq. 23. Note that the $V$ optimization in Eq. 23 consists of two parts: *the maximization of target functional* and *the HJB optimality conditions*. Therefore, we introduce the following loss function by introducing the HJB condition as a regularization term for the target functional:

$$\underbrace{\frac{\lambda_D}{\alpha^{p-1}} \int_0^1 \int_{\mathcal{X}} \left\| \partial_t V_t - \frac{1}{2}\|\nabla V_t\|^2 + \frac{\sigma^2}{2}\Delta V_t \right\|^p d\rho_t dt}_{\text{HJB condition}} - \underbrace{\int_0^1 \int_{\mathcal{X}} \frac{1}{2}\|\nabla V_t\|^2 d\rho_t(x) dt}_{\text{Running cost}}$$

$$\underbrace{- \int_{\mathcal{X}} V_1 d\rho_1 + \int_{\mathcal{X}} \alpha \Psi^* \left( \frac{V_1}{\alpha} \right) d\nu}_{\text{Terminal cost}}, \tag{25}$$

where $1 \le p \le 2$. To achieve a simple parametrization, we set $V = \alpha v_\phi$. Then, up to a constant factor, Eq. 25 can be rewritten as our loss function $\mathcal{L}_\phi$ for $V$ as follows:

$$\mathcal{L}_\phi = \lambda_D \int_0^1 \int_{\mathcal{X}} \left\| \partial_t v_\phi - \frac{\alpha}{2} \|\nabla v_\phi\|^2 + \frac{\sigma^2}{2} \Delta v_\phi \right\|^p d\rho_t dt - \int_0^1 \int_{\mathcal{X}} \frac{\alpha}{2} \|v_\phi\|^2 d\rho_t dt$$
$$- \int_{\mathcal{X}} v_\phi(1, \cdot) d\rho_1 + \int_{\mathcal{X}} \Psi^* \left( v_\phi(1, \cdot) \right) d\nu. \quad (26)$$

**Optimization of Path Measure** $\rho_t$ By introducing the same parametrization $V = \alpha v_\phi$ as $\mathcal{L}_\phi$ into Eq. 24, we also derive the following loss function $\mathcal{L}_\theta$ for $\rho_t$:

$$\mathcal{L}_\theta = \inf_{\rho_0 = \mu} \int_0^1 \int_{\mathcal{X}} \left( \partial_t v_\phi - \frac{\alpha}{2} \|\nabla v_\phi\|^2 + \frac{\sigma^2}{2} \Delta v_\phi \right) d\rho_\theta(t, x). \quad (27)$$

To minimize Eq. 27, it is necessary to obtain the sample $x_t \sim \rho_t$. Here, we exploit the optimality condition for $\rho$, i.e., the reciprocal property, for parametrizing $\rho_t$. Specifically, **we introduce the static generator network** $T_\theta(x, z)$ **for parametrizing** $\mathbb{P}^\star_{1|0}(\cdot|x)$ **(Eq. 16)**, where $z \sim \mathcal{N}(0, I)$ is an auxiliary variable. In other words, the conditional transport plan $\pi^\theta(y|x) := \mathbb{P}^\theta_{1|0}(y|x)$ is parametrized as $\pi^\theta(\cdot|x) = T_\theta(x, \cdot)_{\#}\mathcal{N}(0, I)$. Then, we can simply obtain $x_t$ by leveraging reciprocal property:

$$x_t = (1 - t)x + t\hat{y} + \sigma\sqrt{t(1 - t)}\eta, \quad \hat{y} \sim \pi^\theta(\cdot|x), \quad \eta \sim \mathcal{N}(0, I). \quad (28)$$

Note that this parametrization also provides an additional benefit. By combining the static generator with the reciprocal property, our model offers **simulation-free training**. The previous SB models (Chen et al., 2021a; Shi et al., 2024; Gushchin et al., 2024) typically represent the path measure $\rho_t$ by parametrizing the drift $u$ with neural network in Eq. 3. Therefore, sampling from $\rho_t$ requires SDE or ODE simulation. However, *our model training does not incur simulation costs because it leverages the reciprocal property.*

**Time Discretization** Following other SB algorithms (Chen et al., 2021a; Shi et al., 2024; Gushchin et al., 2024), we discretize the time variable $t$. We simply discretize $t$ into uniform intervals of size $\Delta t = 1/N$, where $N$ denotes the number of timesteps. Then, we approximate $\partial_t v_\phi$ using the following approximation:

$$\partial_t v_\phi(t, x_t) \approx (v_\phi(t + \Delta t, x_{t+\Delta t}) - v_\phi(t, x_t))/\Delta t. \quad (29)$$

Moreover, we introduce the discrete probability distribution for $t$ on the set $\{0, \Delta t, 2\Delta t \dots, (N-1)\Delta t\}$ for the Monte-Carlo estimate of $\mathcal{L}_\phi$ and $\mathcal{L}_\theta$, where $N\Delta t = 1$. The first distribution we consider is the uniform distribution. However, since the terminal cost fluctuates during training, it is advantageous to sample the time variables closer to the terminal time ($t = 1$) more frequently. Thus, we also consider $\mathcal{T}(k\Delta t) = \frac{2(k+1)}{N(N+1)}$ for all $k \in \{0, 1, \dots, N-1\}$. Throughout this paper, we call this distribution as *linear time distribution*.

## 5 EXPERIMENTS

In this section, we evaluate our model on various datasets to assess its performance. In Sec. 5.1, we compare our model's scalability on image datasets with other dynamic OT models. In Sec. 5.2, we test our model on the synthetic datasets to evaluate whether our model learns the correct ESUOT solution. (See Appendix D.1 for the outlier robustness evaluation of the ESUOT problem.) Note that the goal of this paper is to **develop a scalable simulation-free ESUOT algorithm** (Eq. 15). Therefore, in this section, we compare our model with *(1) the dynamical (Entropic) Optimal Transport generative models*, such as Schrödinger Bridge models (Chen et al., 2021a; Shi et al., 2024; Gushchin et al., 2024) and *(2) the dynamic generative models*, such as diffusion models (Song et al., 2021b; Vahdat et al., 2021) and flow matching (Lipman et al., 2023; Tong et al., 2024).

### 5.1 SCALABILITY AND SCRATCH TRAINABILITY ON IMAGE DATASETS

**Scalability Comparison on Generative Modeling with SB Models** Solving the dynamical (entropic) optimal transport is a challenging problem. **The previous SB models (Chen et al., 2021a;**

Table 1: **Image Generation on CIFAR-10.** NFE refers to the Number of Function Evaluations required for sample generation.

| Class | Model | FID ($\downarrow$) | NFE |
|---|---|---|---|
| **Entropic OT (SB)** | **ESUOT-Soft** (Ours) | **3.02** | 1 |
| | **ESUOT-KL** (Ours) | 3.08 | 1 |
| | **EOT** (Ours) | 4.05 | 1 |
| | IPF w/o Pretraining (Chen et al., 2021a) | $\geq 100$ | - |
| | IPF w/ Pretraining (Chen et al., 2021a) | **3.01** | 200 |
| | IMF w/ Pretraining (Shi et al., 2024) | 4.51 | 100 |
| | Gushchin et al. (2024) | $\geq 100$ | - |
| | WLF (Neklyudov et al., 2023) | $\geq 100$ | - |
| **OT** | UOTM (Choi et al., 2023a) | **2.97** | 1 |
| | OTM (Rout et al., 2022) | 7.68 | 1 |
| **Diffusion** | NCSN (Song & Ermon, 2019) | 25.3 | 1000 |
| | DDPM (Ho et al., 2020) | 3.21 | 1000 |
| | Score SDE (VE) (Song et al., 2021b) | 2.20 | 2000 |
| | Score SDE (VP) (Song et al., 2021b) | 2.41 | 2000 |
| | DDIM (50 steps) (Song et al., 2021a) | 4.67 | 50 |
| | CLD (Dockhorn et al., 2022) | 2.25 | 2000 |
| | Subspace Diffusion (Jing et al., 2022) | 2.17 | $\geq 1000$ |
| | LSGM (Vahdat et al., 2021) | 2.10 | 138 |
| **Diffusion Distillation** | CTM (Kim et al., 2024b) | 1.87 | 2 |
| | GDD Distillation (Zheng & Yang, 2024) | **1.54** | 1 |
| **Flow Matching** | FM (Lipman et al., 2023) | 6.35 | 142 |
| | OT-CFM (Tong et al., 2024) | **3.74** | 1000 |

**Shi et al., 2024; Gushchin et al., 2024) exhibit scalability challenges when the distance between the source and target distributions is large.** One example of such a transport problem is generative modeling on an image dataset, where the source is a Gaussian distribution and the target is a high-dimensional image dataset. To address this, the previous SB models (Chen et al., 2021a; Shi et al., 2024) rely on a pretraining strategy, such as the diffusion model with VP SDE (Song et al., 2021b).

In this respect, we evaluate the scalability of our model compared to existing SB models for generative modeling on CIFAR-10 (Krizhevsky et al., 2009). Tab. 1 presents the results. The ESUOT-KL refers to the ESUOT problem with $D_\Psi$=KL divergence and the ESUOT-Soft indicates the ESUOT problem with $\Psi^* = \mathrm{Softplus}$ following (Choi et al., 2023a). For the EOT problem, $\Psi = \iota$, i.e., $\Psi^*(x) = x$ (See Appendix C for details). All other models failed to converge without pretraining procedure, showing FID scores $\geq 100$. In contrast, **our model for the ESUOT problem achieves state-of-the-art results, demonstrating comparable performance to IPF (Chen et al., 2021a) with pretraining.** Our model for the EOT problem also presents an FID score of 4.05, which is a still decent result compared to other models without pretraining. Interestingly, our ESUOT model achieves better distribution matching of the target image distribution, i.e. smaller FID, than our EOT model. This result is interesting because, formally, the ESUOT problem (Eq. 14) allows larger distribution errors in the target data compared to the EOT problem (Eq. 2). We interpret this as a result of the additional optimization benefit of Unbalanced Optimal Transport, as observed in (Choi et al., 2023a). Moreover, **our simulation-free parametrization through the reciprocal property (Eq. 28) offers efficient training and one-step sample generation.** To be more specific, several recently proposed EOT algorithms (Shi et al., 2024; Gushchin et al., 2024) require more than 10 days of training. In contrast, our model takes approximately 4 days to train on 2 RTX 3090Ti GPUs.

**Scalability Comparison on Image-to-image Translation** We evaluated our model on multiple image-to-image (I2I) translation tasks, specifically the *Male → Female* (Liu et al., 2015) ($64 \times 64$) and *Wild → Cat* (Choi et al., 2020) ($64 \times 64$) benchmarks. Here, we tested our SF-ESUOT model for the ESUOT-Soft problem, which performed best in generative modeling on CIFAR-10. Tab. 2 provides the FID scores for these I2I translations tasks. As demonstrated in Tab. 2, our model outperformed other adversarial methods, including other OT-based method of NOT (Korotin et al., 2023), on the Male → Female dataset. Moreover, our model outperformed the dynamic EOT-based methods, such as DSBM (Shi et al., 2024), on the Wild → Cat dataset. These results demonstrate that our model achieves a better approximation of the target distributions in I2I translation tasks, compared to other OT-based approaches. Moreover, Figure 1 illustrates the translated samples on the

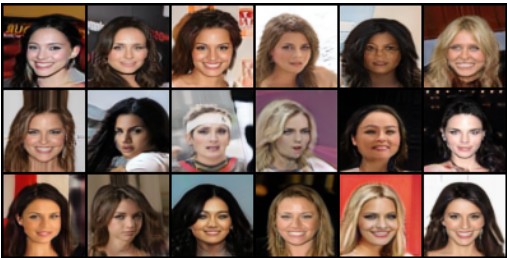 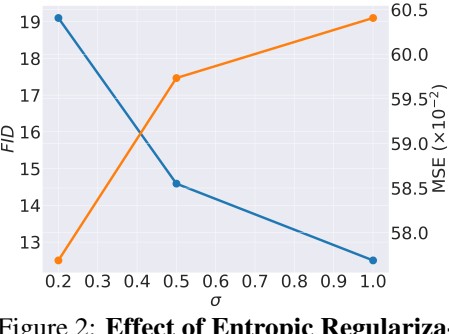

Figure 1: **Unpaired Male → Female translation for 64 × 64 CelebA image.**

Table 2: **FID scores for Image-to-image Translation Tasks. All FID scores of other models are taken from their original paper, except for DSBM, which is taken from De Bortoli et al. (2024).**

| Data | Model | FID |
|------|-------|-----|
| Male→Female (64x64) | DiscoGAN (Kim et al., 2017) | 35.64 |
| | CycleGAN (Zhu et al., 2017) | 12.94 |
| | NOT (Korotin et al., 2023) | 11.96 |
| | **EUOT-Soft** (Ours) | **8.44** |
| Wild→Cat (64x64) | DSBM (Shi et al., 2024) | 25≤ |
| | De Bortoli et al. (2024) | 25≤ |
| | **EUOT-Soft** (Ours) | **14.59** |

Figure 2: **Effect of Entropic Regularization** $\sigma^2$ on Wild→Cat (64x64).

Male → Female dataset. Our model shows strong qualitative performance, faithfully transporting the faces and backgrounds. The additional qualitative results are provided in Appendix D.

**Effect of Entropic Regularization** $\sigma^2$    We performed an ablation study on the entropic regularization intensity $\sigma^2$ of the ESUOT problem (Eq. 14) in the image-to-image translation task on the Wild → Cat (Choi et al., 2020) (64 × 64) dataset. Here, $\sigma$ represents the noise level of the transport dynamics (Eq. 3). Fig. 2 presents the results. As we increase $\sigma$, we observe a consistent trend of decreasing FID scores and increasing transport costs (MSE). The decrease in FID indicates that, with higher $\sigma$, our model better approximates the target distribution. The increase in transport cost (MSE) aligns with our intuition, as $\sigma$ introduces stochasticity into the model. When $\sigma$ becomes too large (e.g., $\sigma = 1$), the transported image may not align well with the identity of the source sample, leading to an increase in transport cost. Conversely, the decrease in FID can be understood as the model is more effectively mapping across diverse images.

## 5.2 COMPARISON TO EOT SOLUTION ON SYNTHETIC DATASET

**Qualitative Results on 2D Toy Datasets**    We evaluate whether **our model can learn the ground-truth solution of the EOT (ESUOT with $\Psi = \iota$) problem.** Specifically, we compare the trained static coupling $\pi_\theta$ (Eq. 14) with the proxy ground-truth coupling obtained using the convex OT solver in the POT library (Flamary et al., 2021). Note that the POT library provides the solution between two *empirical discrete measures* derived from the training data, while our goal is to solve the EOT problem between two *continuous measures* $\mu, \nu$.

Fig. 3 presents the results on two datasets: *Gaussian to 8-Gaussian (G → 8G)* and *Moon to Spiral (M → S)*. (See Appendix C for implementation details). As shown in Fig. 3, our model exhibits a decent performance in learning the EOT transport plan, successfully generating the target distribution and providing a rough approximation of the optimal transport map for individual source samples. However, some noisy outputs are observed, and the model does not fully approximate a precisely optimal transport plan. We interpret this phenomenon in terms of the difficulty in training through PDE-like learning objectives (Li et al., 2024; Luo & Zhou, 2023). In this respect, we believe there is room for development in imposing optimal conditions for the path measure $\rho$. Nevertheless, Sec. 5.1 demonstrates that our model presents considerable scalability enhancement over previous models.

**Quantitative Results on High-Dimensional Gaussian Dataset**    We conduct a quantitative evaluation to test whether our model can accurately learn the optimal coupling. For this evaluation, we

Table 3: **Comparison on Benchmarks on High Dimensional Gaussian Experiments.** $\Delta m$, $\Delta Var$, and $\Delta Cov$ stands for difference of mean, variance, and covariance, respectively. $m$, $Var$, and $Cov$ stands for ground true mean, variance and covariance, respectively.

| Metric (%) ↓ | DSB | IMF-b | DSBM-IPF | DSBM-IMF | RF | Ours |
|---|---|---|---|---|---|---|
| $\Delta m/m$ | 9.3 | 0.9 | 2.3 | 3.5 | 0.4 | 2.9 |
| $\Delta Var/Var$ | 1.22 | ≥ 10 | 0.14 | 0.17 | 8.29 | 3.00 |
| $\Delta Cov/Cov$ | 7.88 | ≥ 10 | 1.63 | 1.52 | - | 3.72 |

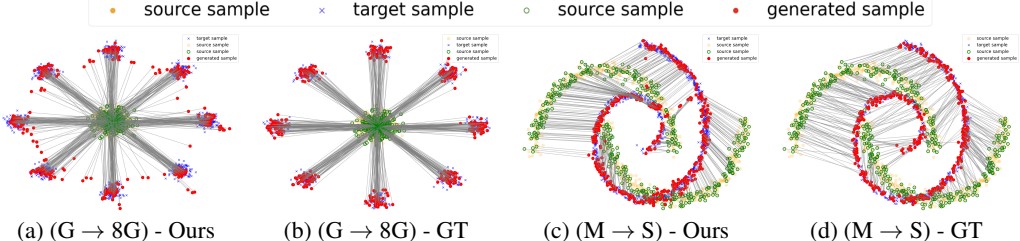

(a) (G → 8G) - Ours     (b) (G → 8G) - GT     (c) (M → S) - Ours     (d) (M → S) - GT

Figure 3: **Comparison of EOT Transport Plan $\pi$ between Our model and Discrete Ground-Truth** from POT (Flamary et al., 2021) when $\sigma = 0.75$. The gray lines illustrate the generated pairs, i.e., the connecting lines between $x$ (green) and $\pi(y|x)$ (red).

exploit the closed form solution for the Entropic Optimal Transport (EOT) problem in the Gaussian-to-Gaussian case (Bunne et al., 2023a). (See the Appendix for the experimental results on the closed-form solution of the ESUOT problem (Janati et al., 2020; Nguyen et al., 2024).) There are two reasons for conduting evaluation on the EOT problem, not on the EUOT problem. First, the EOT problem between Gaussians has a closed-form solution for the optimal transport coupling, while there is no such closed form solution for the EUOT problem. Second, several methods for the Schrödinger Bridge problem, i.e., the dynamic EOT problem, conducted the same benchmark, allowing us to provide broader comparisons with other methods. We follow the experimental settings and evaluation metric of (Shi et al., 2024).

Tab. 3 shows the quantitative evaluation results in terms for three metrics: the relative error of the mean $\Delta m/m$, variance $\Delta Var/Var$, and covariance $\Delta Cov/Cov$ compared to the ground-truth mean, variance, and covariance, respectively. Note that the relative errors for the mean and variance assess the distribution error between the generated distribution $T_{\#}\mu$ and the ground-truth target measure $\nu$. The relative error of covariance evaluates whether our model learned optimal coupling. Overall, our model shows comparable or slightly worse performance than other models. However, since the relative error is below 4% in all three metrics, our model can be considered to exhibit moderate performance. We hypothesize that the adversarial training method and minimizing the PDE-like objective for $T_{\theta}$ resulted in this rough approximation of the optimal coupling. Nevertheless, we would like to emphasize that the strength of our methodology lies in developing a scalable algorithm that transports a source sample into the target with a single-step evaluation. In Sec. 5.1, we discuss further the scalability and efficiency of our proposed method.

# 6 CONCLUSION

In this paper, we propose an algorithm for solving the Entropic Unbalanced Optimal Transport (EUOT) problem. We derived the dynamical formulation of EUOT. Then, we established the dual form and analyzed this dual form from the stochastic optimal control perspective. Our model is based on the simulation-free algorithm leveraging the reciprocal property of the dynamical formulation of EUOT problem. Our experiments demonstrated that our model addresses the scalability challenges of previous Schrödinger Bridge models. Specifically, our model offers simulation-free training and achieves state-of-the-art results in generative modeling on CIFAR-10 without diffusion model pretraining. A limitation of this work is that our method demonstrates lower accuracy in learning the EUOT compared to other models. We hypothesize that this is due to the inherent difficulty of achieving precise matching using a PINN-style loss function. Additionally, due to computational resource constraints, we were unable to test our model on high-resolution datasets such as CelebA-HQ (Liu et al., 2015) ($256 \times 256$).

## ETHICS STATEMENT

Our approach significantly enhances the scalability of EOT algorithms, enabling the generation of high-quality samples from large-scale datasets while maintaining an accurate representation of the data distribution. As a result, we expect our model to impact various fields, including image transfer, finance, image synthesis, healthcare, and anomaly detection. However, it is important to recognize the potential negative societal implications of our work. Generative models can unintentionally learn and magnify existing biases within the data, which may reinforce societal biases. Therefore, careful monitoring and control are crucial when deploying these models in real-world applications. Rigorous management of both the training data and the modeling process is essential to mitigate any potential negative societal effects.

## REPRODUCIBILITY STATEMENT

To ensure the reproducibility of our work, we submitted the anonymized source in the supplementary material, provided complete proofs of our theoretical results in Appendix A, and included the implementation and experiment details in Appendix C.

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

## A PROOFS AND DERIVATIONS

In this section, we provide the proof of the theorems in Sec. 3. Moreover, we introduce another dual form and its relationship to our work. Furthermore, we derive the optimization problem Eq. 24 and justify the conditional sampling in line 5 of Algorithm 1. For all theorems, we assume that $\Psi$ is an differentiable entropy function that satisfies superlinearity, i.e. $\Psi'_\infty := \lim_{x\to\infty} \Psi(x)/x = \infty$. In this case, $D_\Psi(\rho_1|\nu)$ is infinity whenever $\rho_1$ has singularity with respect to $\nu$. Thanks to the superlinearity, continuity and convexity of $\Psi$, $D_\Psi$ is a lower semi-coninituous function.

### A.1 PROOFS

The following lemma implies that the search space of the joint distribution $\pi$ in the EUOT problem can be extended to unnormalized density space $\mathcal{M}_2$. Based on this lemma, we abuse the notation for the search space in the following theorems.

**Lemma A.1.** *Let $\pi^\star$ be the optimal plan for*

$$\inf_{\pi_0=\mu, \pi\in\mathcal{M}_2(\mathcal{X}\times\mathcal{X})} \left[ \int_{\mathcal{X}\times\mathcal{X}} \frac{1}{2}\|x-y\|^2 d\pi(x,y) - \sigma^2 H(\pi) + \alpha D_\Psi(\pi_1|\nu) \right]. \tag{30}$$

*Note that the search space of $\pi$ is extended to unnormalized density space $\mathcal{M}_2$ instead of using $\mathcal{P}_2$ as in EUOT problem defined in Eq. 14. Even if the search space is extended, the mass of the optimal target marginal $\pi_1^\star$ is 1. In other words, the problem Eq. 30 is equivalent to Eq. 14.*

*Proof.* The well-known dual form of the Eq. 30 Genevay (2019) is defined as follows:

$$\sup_{u,v} \int_{\mathcal{X}} u(x)d\mu(x) - \int_{\mathcal{X}} \Psi^*(-v(y))d\nu(y) - \epsilon \int_{\mathcal{X}\times\mathcal{X}} e^{\frac{u(x)+v(y)-c(x,y)}{\epsilon}} d\mu(x)d\nu(y). \tag{31}$$

Thanks to the Fenchel-Rockafellar theorem Singer (1979), the strong duality holds (See Proposition 4.2 in Séjourné et al. (2022). The first variation of Eq. 31 with respect to the pair of the optimal potentials $(u^\star, v^\star)$ is as follows:

$$\int_{\mathcal{X}} \delta u(x)d\mu(x) + \int_{\mathcal{X}} \delta v \Psi^{*'}(-v^\star(y))d\nu(y) - \int_{\mathcal{X}\times\mathcal{X}} (\delta u + \delta v) e^{\frac{u^\star(x)+v^\star(y)-c(x,y)}{\epsilon}} d\mu(x)d\nu(y). \tag{32}$$

Now, let $\tilde{\nu}(y) = \Psi^{*'}(-v^\star(y))\nu(y)$. If the $(\delta u, \delta v) = (\lambda, -\lambda)$, then the Eq. 32 can be written as follows:

$$\int \lambda d\mu - \int \lambda d\tilde{\nu} = \lambda(1 - m(\tilde{\nu})), \tag{33}$$

where $m(\cdot)$ denotes the mass of the measure. Since the potentials are optimal, the mass of $\tilde{\nu}$ should be 1. Furthermore, reordering the first variation with respect to $\delta v$ in Eq. 32, we can derive that

$$\Psi^{*'}(-v^\star(y)) = \int e^{\frac{u^\star(x)+v^\star(y)-c(x,y)}{\epsilon}} d\mu(x). \tag{34}$$

Then, by leveraging the primal-dual relationship Séjourné et al. (2022), i.e.

$$d\pi^\star(x,y) = e^{\frac{u^\star(x)+v^\star(y)-c(x.y)}{\epsilon}} d\mu(x)d\nu(y), \tag{35}$$

we can derive the following equation:

$$\pi_1^\star(y) = \int \pi^\star(x,y)dx = \left( \int e^{\frac{u^\star(x)+v^\star(y)-c(x.y)}{\epsilon}} d\mu(x) \right) \nu(y) = \Psi^{*'}(-v^\star(y))\nu(y) = \tilde{\nu}(y). \tag{36}$$

Since $m(\tilde{\nu}) = 1$, $\pi_1^\star$ has a mass of 1. □

**Theorem A.2.** *The EUOT problem is equivalent to the following dynamic transport problem:*

$$\inf_u \left[ \int_0^1 \int_{\mathcal{X}} \frac{1}{2}\|u_t(x)\|^2 d\rho_t(x)dt + \alpha D_\Psi(\rho_1|\nu) \right], \tag{37}$$

*where $\partial_t\rho_t + \nabla\cdot(u\rho_t) - \frac{\sigma^2}{2}\Delta\rho_t = 0$ and $\rho_0 = \mu$. Moreover, the optimal solution $\mathbb{P}^\star$ satisfies the reciprocal property, i.e.,*

$$\mathbb{P}_t^\star(\cdot|x,y) = \mathcal{N}(\cdot|(1-t)x + ty, \sigma^2 t(1-t)I), \quad (x,y) \sim \mathbb{P}_{0,1}^\star \text{ almost surely.} \tag{38}$$

*Proof.* Suppose $u^\star$ is the solution of Eq. 37. Let $\mathbb{P}^\star$ be the path measure induced by $X^{u^\star}$. Since $\Psi'_\infty = \infty$, $\mathbb{P}^\star_1(= \rho_1)$ is absolutely continuous with respect to $\nu$. Note that $\mathbb{P}^\star_1 \in \mathcal{P}_2(\mathcal{X})$ by the Lemma A.1. This implies that the objective of $u$ in Eq. 37 is to solve the SB problem between $\mu$ and $\mathbb{P}^\star_1$. Thus, $\mathbb{P}^\star_t$ satisfies the reciprocal property. Moreover, using the reciprocal property, Eq. 37 could be reformulated to the static formulation:

$$\inf_{\pi \in \mathcal{P}_2,\ \pi_0 = \mu} \left[ \sigma^2 D_{\mathrm{KL}}(\pi | \mathbb{Q}_{0,1}) + \alpha D_\Psi(\pi_1 | \nu) \right]. \tag{39}$$

Now, by applying Eq. 10, we obtain

$$\inf_{\pi_0 = \mu, \pi \in \mathcal{P}_2(\mathcal{X})} \left[ \int_{\mathcal{X} \times \mathcal{X}} \frac{1}{2} \|x - y\|^2 d\pi(x,y) - \sigma^2 H(\pi) + \alpha D_\Psi(\pi_1 | \nu) \right]. \tag{40}$$

$\square$

**Proposition A.3** (**Dual I**). *The EUOT problem is equivalent to the following problem:*

$$\sup_{V \in \Phi_{2,b}} \inf_u \left[ \mathbb{E}\left[ \int_0^1 \frac{1}{2} \|u_t(X^u_t)\|^2 dt + V(X^u_1) \right] - \int_{\mathcal{X}} \alpha \Psi^* \left( \frac{V(y)}{\alpha} \right) d\nu(y) \right], \quad \text{s.t. } X_0 \sim \mu. \tag{41}$$

*Proof.* Let $F(\pi) := \sigma^2 D_{\mathrm{KL}}(\pi | \mathbb{Q}_{0,1})$ and $G(\pi) := D_\Psi(\pi_1 | \nu)$. Then, Eq. 39 can be rewritten as follows:

$$\inf_{\pi \in \mathcal{P}_2} [F(\pi) + G(\pi)]. \tag{42}$$

Note that $F$ and $G$ are convex lower semi-continuous functions. Thus, by applying Fenchel-Rockafellar theorem Singer (1979), we obtain the following duality form:

$$\inf_{\pi \in \mathcal{P}_2} F(\pi) + G(\pi) = \sup_{V \in \mathcal{P}_2^*} [-F^*(-V) - G^*(V)]. \tag{43}$$

Note that $\mathcal{P}_2^* = \Phi_{2,b}$ by Lemma 9.8 in Gozlan et al. (2017). Moreover, by proof of Theorem 9.5 in Gozlan et al. (2017),

$$-F^*(-V) = \inf_{\pi \in P_2(\mathcal{X})} \left[ \sigma^2 D_{\mathrm{KL}}(\pi | \mathbb{Q}_{0,1}) + \int V(y) d\pi_1(y) \right]. \tag{44}$$

Since $G^*(V) = \int \alpha \Psi^*(V(y)/\alpha) d\nu(y)$, we finally obtain the following dual form:

$$\sup_{V \in \Phi_{2,b}} \inf_{\pi \in \mathcal{P}_2} \left[ \sigma^2 D_{\mathrm{KL}}(\pi | \mathbb{Q}_{0,1}) + \int V(y) d\pi_1(y) - \int_{\mathcal{X}} \alpha \Psi^* \left( \frac{V(y)}{\alpha} \right) d\nu(y) \right]. \tag{45}$$

Through the discussion in the proof of Theorem A.2 or Sec. 2.2 in Gushchin et al. (2024), $\pi$ can be replaced by the distribution of the stochastic process $\{X^u_t\}$ and $D_{\mathrm{KL}}(\pi | \mathbb{Q}_{0,1}) = \mathbb{E}\left[\int \|u_t\|^2/(2\sigma^2) dt\right]$. By the replacement, we obtain the following dual form of EUOT.

$$\sup_{V \in \Phi_{2,b}} \inf_u \left[ \mathbb{E}\left[ \int_0^1 \frac{1}{2} \|u_t(X^u_t)\|^2 dt + V(X^u_1) \right] - \int_{\mathcal{X}} \alpha \Psi^* \left( \frac{V(y)}{\alpha} \right) d\nu(y) \right], \tag{46}$$

where $X_0 \sim \mu$. $\square$

## A.2 Derivations

We provide the another dual formulation of dynamical EUOT. Then, we also introduce the connection to our dual form, i.e. Eq. 17. Furthermore, we derive the optimization problem Eq. 24. Finally, we provide the justification of conditional sampling in line 5 of Algorithm 1.

**Lagrangian Dual of Dynamical Form of EUOT**    Starting from the dynamical formulation of EUOT (Eq. 37), we can also derive an alternative dual formulation of EUOT. Note that when $\Psi$ is a convex indicator, the following Eq. 47 corresponds to the SB objective of Neklyudov et al. (2023) (Example 4.4). Specifically, the dual formulation is expressed as follows:

**Proposition A.4** (**Dual II**). *The dual form of Eq. 37 is written as follows:*

$$\sup_A \inf_\rho \left[ \int_0^1 \int_{\mathcal{X}} \left( \partial_t A_t - \frac{1}{2} \|\nabla A_t\|^2 + \frac{\sigma^2}{2} \Delta A_t \right) d\rho_t dt + \int_{\mathcal{X}} A_0 d\mu - \int_{\mathcal{X}} \alpha \Psi^* \left( \frac{A_1}{\alpha} \right) d\nu \right], \quad (47)$$

*where $\rho_0 = \mu$ and $A : [0,1] \times \mathcal{X} \to \mathbb{R}$. Furthermore, for the optimal $u^\star$ in Eq. 37 and optimal $A^\star$, $u^\star = -\nabla A^\star$.*

*Proof.* The dual form of Eq. 37 is

$$\inf_{\rho,u} \sup_{A,\lambda_0,\lambda_1} \int_0^1 \int_{\mathcal{X}} \frac{1}{2} \|u_t(x)\|^2 \rho_t(x) dx dt + \alpha \int_{\mathcal{X}} \Psi \left( \frac{dP(x)}{d\nu(x)} \right) d\nu(x)$$
$$- \int_0^1 \int_{\mathcal{X}} A(t,x) \left( \partial_t \rho_t(x) + \nabla \cdot (u_t \rho_t(x)) - \frac{\sigma^2}{2} \Delta \rho_t(x) \right) dx dt \quad (48)$$
$$+ \int_{\mathcal{X}} \lambda_0(x)(\rho_0(x) - \mu(x)) dx + \int_{\mathcal{X}} \lambda_1(x)(\rho_1(x) - P(x)) dx,$$

where $A : [0,1] \times \mathcal{X} \to \mathbb{R}$ and $\lambda_0, \lambda_1 : \mathcal{X} \to \mathbb{R}$ are the dual variables. Note that we can freely swap the order of the optimization since the optimization problem is convex in $\rho$, $P$, $u$, and linear in $A$, $\lambda_0$, $\lambda_1$. By applying integration by parts to second line of Eq. 48, we obtain

$$\sup_{A,\lambda_0,\lambda_1} \inf_{\rho,P,u} \int_0^1 \int_{\mathcal{X}} \frac{1}{2} \|u_t\|^2 d\rho_t dt + \alpha \int_{\mathcal{X}} \Psi \left( \frac{dP}{d\nu} \right) d\nu$$
$$+ \int_0^1 \int_{\mathcal{X}} \left( \partial_t A_t + \langle u_t, \nabla A_t \rangle + \frac{\sigma^2}{2} \Delta A_t \right) d\rho_t dt - \int_{\mathcal{X}} A_1 d\rho_1 + \int_{\mathcal{X}} A_0 d\rho_0 \quad (49)$$
$$+ \int_{\mathcal{X}} \lambda_0(x)(\rho_0(x) - \mu(x)) dx + \int_{\mathcal{X}} \lambda_1(x)(\rho_1(x) - P(x)) dx.$$

Here, note that the last two terms in the second line of Eq. 48 pop out from the integration of parts with respect to the time variable $t$. Since the above problem with respect to $u$ is quadratic, we can obtain the explicit solution $u = -\nabla A$. Moreover, by freely swapping the optimization variables, we obtain $\lambda_0 = A_0$, $\lambda_1 = A_1$, $\rho_0 = \mu$, and $\rho_1 = P$. Moreover, by the definition of convex conjugate, we obtain Eq. 47:

$$\sup_A \inf_\rho \int_0^1 \int_{\mathcal{X}} \partial_t A_t - \frac{1}{2} \|\nabla A_t\|^2 + \frac{\sigma^2}{2} \Delta A_t \, d\rho_t dt$$
$$- \int_{\mathcal{X}} A_1 \frac{d\rho_1}{d\nu} - \alpha \Psi \left( \frac{d\rho_1}{d\nu} \right) d\nu + \int_{\mathcal{X}} A_0 d\mu, \quad (50)$$
$$= \inf_\rho \sup_A \int_0^1 \int_{\mathcal{X}} \left( \partial_t A_t - \frac{1}{2} \|\nabla A_t\|^2 + \frac{\sigma^2}{2} \Delta A_t \right) d\rho_t dt - \int_{\mathcal{X}} \alpha \Psi^* \left( \frac{A_1}{\alpha} \right) d\nu + \int_{\mathcal{X}} A_0 d\mu. \quad (51)$$

$\square$

**Relationship between Two Dual Formulations** We establish the equivalence between the two dual forms: *Dual I* and *Dual II*.

**Proposition A.5.** *The Lagrangian dual formulation of Eq. 41 (Dual I) is equivalent to Eq. 47 (Dual II). Moreover, let $V^\star : \mathcal{X} \to \mathbb{R}$ be the solution of $V$ in Eq. 41 and let $A^\star : [0,1] \times \mathcal{X} \to \mathbb{R}$ and $\rho^\star$ be the solution of $A$ and $\rho$ in Eq. 47, respectively. Then, $V^\star(x) = A_1^\star(x)$ in $\rho_1^\star$ almost surely.*

*Proof.* We derive the results by following the proof of Proposition A.4. The Lagrangian dual of Eq. 41 can be reformulated as follows:

$$\sup_{V,A} \inf_{u_t,\rho} \int_0^1 \int_{\mathcal{X}} \frac{1}{2}\|u\|^2 d\rho_t dt + \int_{\mathcal{X}} V_1 d\rho_1 - \int_{\mathcal{X}} \alpha\Psi^*\left(\frac{V_1}{\alpha}\right) d\nu$$

$$- \int_0^1 \int_{\mathcal{X}} A\left[\partial_t\rho + \nabla\cdot(u_t\rho) - \frac{\sigma^2}{2}\Delta\rho\right] dxdt, \quad (52)$$

$$= \sup_{V,A} \inf_{\rho\in\mathcal{M}_2} \int_{\mathcal{X}} V_1 - A_1 d\rho_1 + \int_{\mathcal{X}} A_0 d\rho_0 - \int_{\mathcal{X}} \alpha\Psi^*\left(\frac{V_1}{\alpha}\right) d\nu$$

$$+ \int_0^1 \int_{\mathcal{X}} \left(\partial_t A_t - \frac{1}{2}\|A_t\|^2 + \frac{\sigma^2}{2}\Delta A_t\right) d\rho_t dt, \quad (53)$$

where $\rho_0 = \mu$. Note that $\rho$ is a non-negative Borel measure. If $V_1(x) < A_1(x)$ for some $x$, then the infimum of the first term of the above equation with respect to $\rho_1$ becomes $-\infty$. Thus, $V_1 \geq A_1$. Moreover, whenever $V_1 > A_1$, the corresponding $\rho_1$ vanishes. Thus, $V_1^\star = A_1^\star$ for $\rho_1^\star$-almost surely. Therefore, by the optimality condition, the problem boils down to the following optimization problem:

$$\sup_A \inf_\rho \int_{\mathcal{X}} A_0 d\rho_0 - \int_{\mathcal{X}} \alpha\Psi^*\left(\frac{A_1}{\alpha}\right) d\nu + \int_0^1 \int_{\mathcal{X}} \left(\partial_t A_t - \frac{1}{2}\|A_t\|^2 + \frac{\sigma^2}{2}\Delta A_t\right) d\rho_t dt. \quad (54)$$

$\square$

**Derivation of Inner-loop Objective (Eq. 24)** The inner-loop problem of Eq. 22 can be written as follows:

$$\inf_{(u,\rho)} \mathbb{E}_{(t,x)\sim\rho}\left[\frac{1}{2}\|u_t(x)\|^2\right] + \mathbb{E}_{\hat{y}\sim\rho_1}\left[V(1,\hat{y})\right],$$

$$\text{s.t. } \partial_t\rho + \nabla\cdot(u\rho) - (\sigma^2/2)\Delta\rho = 0, \; \rho_0 = \mu. \quad (55)$$

Then, the dual form of Eq. 55 with the dual variable $\lambda : [0,1] \times \mathcal{X} \to \mathbb{R}$ can be easily derived as follows:

$$\sup_\lambda \inf_{(u,\rho)} \int_{\mathcal{X}} \int_0^1 \frac{1}{2}\|u_t(x)\|^2 \rho(t,x)dtdx + \int_{\mathcal{X}} V(1,\hat{y})\rho(1,\hat{y})$$

$$- \int_{\mathcal{X}} \int_0^1 \lambda(t,x)\left(\partial_t\rho(t,x) + \nabla\cdot(u_t\rho) - \frac{\sigma^2}{2}\Delta\rho\right) dtdx, \quad (56)$$

where $\rho_0 = \mu$. From now on, we omit the condition $\rho_0 = \mu$ for simplicity. By applying integration by parts to the last term of Eq. 56, we obtain

$$\sup_\lambda \inf_{(u,\rho)} \int_{\mathcal{X}} \int_0^1 \frac{1}{2}\|u_t(x)\|^2 \rho(t,x)dtdx + \int_{\mathcal{X}} (V(1,\hat{y}) - \lambda(1,\hat{y}))\rho(1,\hat{y}) + \int_{\mathcal{X}} \lambda(0,x)d\mu(x)$$

$$+ \int_{\mathcal{X}} \int_0^1 \left(\partial_t\lambda(t,x) + \nabla\lambda(t,x)\cdot u_t(x) - \frac{\sigma^2}{2}\Delta\lambda(t,x)\right) \rho(t,x)dtdx. \quad (57)$$

Since the minimization objective of Eq. 57 is quadratic with respect to variable $u$, we can derive the explicit solution $u = -\nabla\lambda$. Thus, we obtain

$$\sup_\lambda \inf_\rho \left[\int_0^1 \int_{\mathcal{X}} \left(\partial_t\lambda - \frac{1}{2}\|\nabla\lambda\|^2 + \frac{\sigma^2}{2}\Delta\lambda\right) \rho_t dxdt + \int_{\mathcal{X}} (V_1 - \lambda_1)\rho_1 dx + \int_{\mathcal{X}} \lambda_0 d\mu\right]. \quad (58)$$

Finally, using the similar argument as the proof of Proposition A.5, we obtain $\lambda = V$ $\rho_t^\star$-a.s. and consequently, $u = -\nabla V$. Finally, by substituting $V$ into $\lambda$, we obtain

$$\inf_\rho \left[\int_0^1 \int_{\mathcal{X}} \left(\partial_t V - \frac{1}{2}\|\nabla V\|^2 + \frac{\sigma^2}{2}\Delta V\right) \rho_t dxdt\right] + \int_{\mathcal{X}} V_0 d\mu. \quad (59)$$

**Conditional Sampling** In this paragraph, we justify the sampling $x_{t+\Delta t} \sim \mathbb{P}_{t,1}(\cdot|x_t, \hat{y})$ in line 5 of Algorithm 1. Notice that we are sampling with the assumption of reciprocal property:

$$\mathbb{P}_{t+\Delta t|t,1}(\cdot|x_t, \hat{y}) = \mathbb{Q}_{t+\Delta t|t,1}(\cdot|x_t, \hat{y}). \tag{60}$$

Since $\mathbb{Q}_{t,1}$ is a Gaussian distribution of variance $\sigma^2(1-t)$, the variance of $\mathbb{Q}_{t+\Delta t|t,1}$ is

$$\sigma^2(1-t) \times \frac{\Delta t}{1-t} \times \frac{1-t-\Delta t}{1-t} = \sigma^2 \frac{\Delta t(1-t-\Delta t)}{(1-t)^2}.$$

Then, the average of $x_{t+\Delta t}$ is

$$\mathbb{P}_{t+\Delta t|t,1}(x_{t+\Delta t}|x_t, \hat{y}) = \mathcal{N}\left(x_{t+\Delta t}\left|\frac{\Delta t}{1-t}x_t + \frac{1-t-\Delta t}{1-t}\hat{y}, \ \sigma^2 \frac{\Delta t(1-t-\Delta t)}{(1-t)^2}\right.\right). \tag{61}$$

# B CONNECTION TO RELATED WORKS

In this section, we clarify the connection of our method to various existing OT algorithms.

## B.1 CONNECTION TO WASSERSTEIN LAGRANGIAN FLOW

In this section, we clarify the relationship of our approach compared to Neklyudov et al. (2023). In Neklyudov et al. (2023), they suggests a general framework for handling dynamical optimal transport problems for various cost functionals where the marginal distributions are fixed for some timesteps $\{0 = t_0, \ldots, t_{N-1}, t_N = 1\}$. The algorithm is derived by leveraging the Lagrangian dual formulations of these problems. Technically, this algorithm alternately updates the probability density $\rho_t$ and value function $V$. It is easy to show that when the methodology proposed in Neklyudov et al. (2023) is reduced to our problem, it becomes a max-min problem of Dual II (Proposition A.4).

In our approach, we similarly derive a the objective for the density $\rho$ Appendix A.2 by leveraging the lagrangian dual formulation. The key difference to the objective derived in Neklyudov et al. (2023) lies in our active incorporation of optimality conditions of our EUOT formulation, which allowed us to avoid the minimax objective on the HJB-like equation (Eq. 47). We believe this difference contributes to the scalability of our network, though the further investigation in required to fully understand its impact.

Furthermore, there is a significant difference in the evaluation process. Unlike our approach, Neklyudov et al. (2023) does not directly learn the transport map. Instead, they train the value function $v_\phi$ and generate samples by solving stochastic differential equation (SDE) dynamics using the gradient of $v_\phi$, requiring approximately 100 function evaluations (NFE) for the evaluation. In contrast, our method directly parametrizes the transport map, enabling one-step evaluation.

## B.2 CONNECTION TO DIFFUSION SCHRÖDINGER BRIDGE MATCHING (DSBM)

A key difference between our method and Schrödinger bridge matching Shi et al. (2024); Liu et al. is that our approach eliminates burdensome simulations by directly parametrizing the transport plan $T_\theta$. In Shi et al. (2024), the drift $u := u_\theta$ of the SDE $dX_t^u = udt + \sigma dW_t$ is learned in the following way: first, sample pairs $\{(X_0^u, X_1^u)\}$ are obtained through SDE simulation. Then, sample an intermediate samples using the reciprocal property $X_t \sim \rho_{t|0,1}(\cdot|X_0^u, X_1^u)$. Note that this approach requires the heavy simulation of $X_1^u$ starting from $X_0^u$. Then, the drift $u_\theta$ is updated by Markovian projection (or action matching) using these triplets $\{(X_0^u, X_t, X_1^u)\}$.

In contrast, our model directly obtain $\{(X_0, X_1)\}$ through the transport plan $T_\theta$. Then, we directly obtain $X_t \sim \rho_{t|0,1}(\cdot|X_0, X_1)$ by leveraging reciprocal property (Eq. 16). While our method reduces the simulation burden, it face some computational burden by the PINN-like objective.

## B.3 OTHER RELATED WORKS

In this paragraph, we introduce several works that approximate the solution of EUOT between the continuous measures. It is challenging to directly address the EUOT problem in a continuous setting Gazdieva et al. (2023). Hence, several approaches utilize discrete minibatch EUOT approximations

based on Sinkhorn Algorithm Eyring et al. (2024); Klein et al. (2023). These works employ minibatch EUOT plans to train flow matching Lipman et al. (2023) or conditional flow matching Tong et al. (2023). On the other hand, Gazdieva et al. (2023) proposes a lightweight solver for the EUOT problem. This method is based on a Gaussian mixture approximation of the EUOT plan. These approaches rely on strong assumptions that the OT plan can be approximated by a minibatch EUOT plan or a Gaussian mixture. To the best of our knowledge, our work is the first attempt to propose a method for solving the continuous EUOT problem in Eq. 14, without any additional assumptions on the EUOT plan. Notably, our method also benefits from simulation-free training and offers one-step sample generation.

## C IMPLEMENTATION DETAILS

Unless otherwise stated, the source distribution $\mu$ is a standard Gaussian distribution with the same dimension as the data (target) distribution $\nu$. Moreover, computing $\Delta v_\phi$ in line 6 is burdensome, hence, we approximate it through Hutchinson-Skilling trace estimator Hutchinson (1989); Skilling (1989). Our implementation of the trace estimator follows the likelihood estimation implemented in Song et al. (2021b).

### C.1 2D EXPERIMENTS

**Data Description** We obtain each data as follows:

- **8-Gaussian**: For $m_i = 12 \left( \cos \frac{i}{4}\pi, \sin \frac{i}{4}\pi \right)$ for $i = 0, 1, \ldots, 7$ and $\sigma = 0.04$, the target distribution is defined as the mixture of $\mathcal{N}(m_i, \sigma^2)$ with an equal probability.
- **Moon to Spiral**: We follow Choi et al. (2024).
- **Gaussian Experiments**: We follow Shi et al. (2024), note that the data dimension is 50.

**Network Architectures** Let $y$ be concatenated variable of source data $x$ and auxiliary variable $z \sim \mathcal{N}(0, I)$. The dimension of the auxiliary variable is set to be same as the dimension of data. We parametrize the generator by $T_\theta(x, z) = x + t_\theta(y)$ where $t_\theta$ is the MLP of three hidden layers. Here, $y$ is the concatenation of $x$ and $z$. Moreover, for the discriminator, we concatenate $x$ and $t$ and pass it through a 3-layered MLP. We employ a hidden dimension of 256 and a SiLU activation function.

**Training Hyperparameters** We trained for 120K iterations with the batch size of 1024. We set Adam optimizer with $(\beta_1, \beta_2) = (0, 0.9)$, learning rate of $2 \times 10^{-4}$ and $10^{-4}$ for the transport map $T_\theta$ and the potential $v_\phi$, respectively. We use a cosine scheduler to gradually decrease the learning rate from $10^{-4}$ to $5 \times 10^{-5}$. We update the inner objective (the generator) three times for every single update of the outer objective. We use the number of timesteps of 20, $\alpha = 1$, and uniform distribution for $\mathcal{T}$. Moreover, we set $\lambda_G = 0.1$, $\lambda_D = 1$, $p = 1$ and $\sigma = 0.8$.

**Discrete OT Solver** We used the POT library Flamary et al. (2021) to obtain an accurate transport plan $\pi_{pot}$. We used 1024 training samples for each dataset in estimating $\pi_{pot}$ to sufficiently reduce the gap between the true continuous measure and the empirical measure.

### C.2 IMPLEMENATION ON IMAGE DATASETS

**CIFAR-10 Generation** We follow the generator $T_\theta$ architecture and the implementation of Choi et al. (2023a). Moreover, we follow the discriminator architecture in Choi et al. (2023b). We trained for 300K iterations with the batch size of 256. We use Adam optimizer with $(\beta_1, \beta_2) = (0, 0.9)$, learning rate of $10^{-4}$. We use a cosine scheduler to gradually decrease the learning rate from $10^{-4}$ to $5 \times 10^{-5}$. We used a gradient clip of 1.0 for both the generator and discriminator. We use the number of timesteps of 20, $\alpha = 1$, $p = 2$, $\sigma = 0.1$, and linear time distribution $\mathcal{T}$. Moreover, when $\Psi^*(x) = 2\log(1 + e^x) - 2\log 2$ (**EUOT-Soft**), we set $\lambda_G = 5$, $\lambda_D = 1$. When $\Psi^*(x) = 5e^{x/5} - 5$ (**EUOT-KL**), we use $\lambda_G = 1$, $\lambda_D = 1$. Note that $\Psi^*(x) = 5e^{x/5} - 5$ is the convex conjugate of entropy function $\Psi$ for $5D_{\mathrm{KL}}$. Unless otherwise stated, we consider $\Psi^*(x) = 2\log(1 + e^x) - 2\log 2$. To ensure stable training, we introduce the $R_1$ regularization term with a regularization coefficient of

$\lambda = 2.5$ during the update of the potential $v_\phi$. n other words, we add $\lambda\|\nabla v_\phi(t,x)\|^2$ on line 8 of Alg, 1.

**Image-to-image Translation**    We follow the generator $T_\theta$ architecture of Choi et al. (2023a). For the discriminator, we follow the largest discriminator used in Choi et al. (2023a) when training CelebA-HQ 256x256 dataset. We train for 30K iterations with the batch size of 64. We use Adam optimizer with $(\beta_1, \beta_2) = (0, 0.9)$. The learning rate of generator and discriminator is $2 \times 10^{-4}$ and $10^{-4}$, respectively. We use the number of timesteps of 20, $\alpha = 1$, $p = 2$, $\sigma = 0.5$, $\lambda_G = \lambda_D = 1$, and uniform time distribution $\mathcal{T}$. Note that in the image translation experiments, we do not use clip nor $R_1$ regularization.

**Evaluation Metric**    For the evaluation of CIFAR10, we used 50,000 generated samples to measure FID (Karras et al., 2019) scores. For the Wild→Cat experiments, we compute FID statistics based on train dataset of the target dataset. Then, we generate samples $T_\theta(x)$ from each source sample $x$ from test dataset. We sampled multiple generated samples from each source. Then, we calculate the FID based on this generated samples. Note that this follows the implementation of De Bortoli et al. (2024). For CelebA experiment, we follow the metric used in Korotin et al. (2023). Specifically, we compute FID through test datasets.

## D    ADDITIONAL RESULTS

### D.1    OUTLIER ROBUSTNESS OF SF-ESUOT ALGORITHM

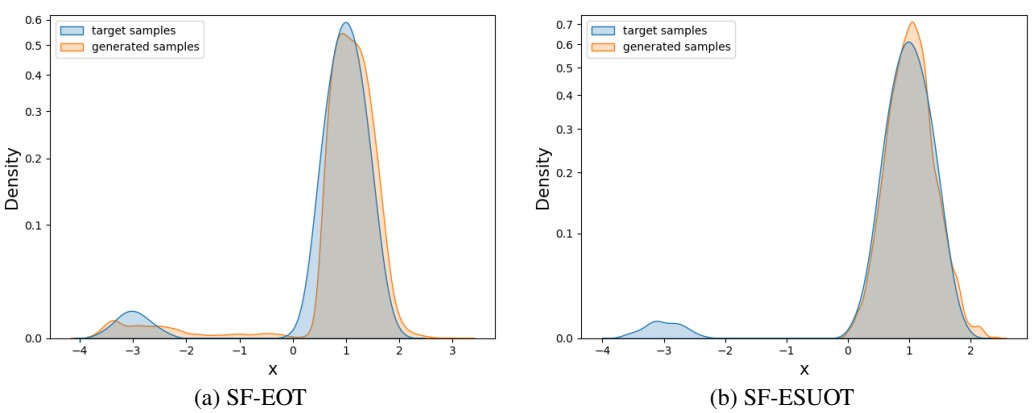

(a) SF-EOT                                    (b) SF-ESUOT

Figure 4: Outlier robustness test on the synthetic dataset with $2\%$ outlier. Each subfigure visualizes the target density $\nu$ and the generated density $\pi_1$.

**Experimental Settings**    The unbalanced version of the (E)OT problem offers outlier robustness by relaxing the marginal distribution constraint (Séjourné et al., 2022; Balaji et al., 2020; Choi et al., 2023a). Following (Choi et al., 2023a), we evaluated the outlier robustness of the Entropic Semi-Unbalanced Optimal Transport (ESUOT) problem using a synthetic dataset containing $2\%$ outliers. Specifically, we tested our SF-ESUOT model and the SF-EOT model (our model for the EOT problem without unbalancedness). Both models are trained for 20K iterations with a batch size of 1024. We set $\alpha = 10^{-4}$, $\sigma = 0.1$, $p = 1$, $\lambda_G = 1$, and $\lambda_D = 0.1$. All other training hyperparameters are the same as those used in the 2D experiments (Sec C.1).

**Experimental results**    The results are demonstrated in Fig 4. Because the EOT problem assumes the exact matching of the target distribution, the SF-EOT model tries to match both in-distribution and outlier densities. However, this leads to the undesirable behavior of generating density outside the target support (around $x = -1$). In contrast, the SF-ESUOT problem focuses on matching the in-distribution density, demonstrating outlier robustness. This robustness arises from the relaxation of the marginal distribution constraint in the ESUOT problem. This outlier robustness demonstrates the importance of investigating the ESUOT problem, because many real-world applications are exposed to diverse outlier samples.

## D.2 ADDITIONAL QUALITATIVE RESULTS

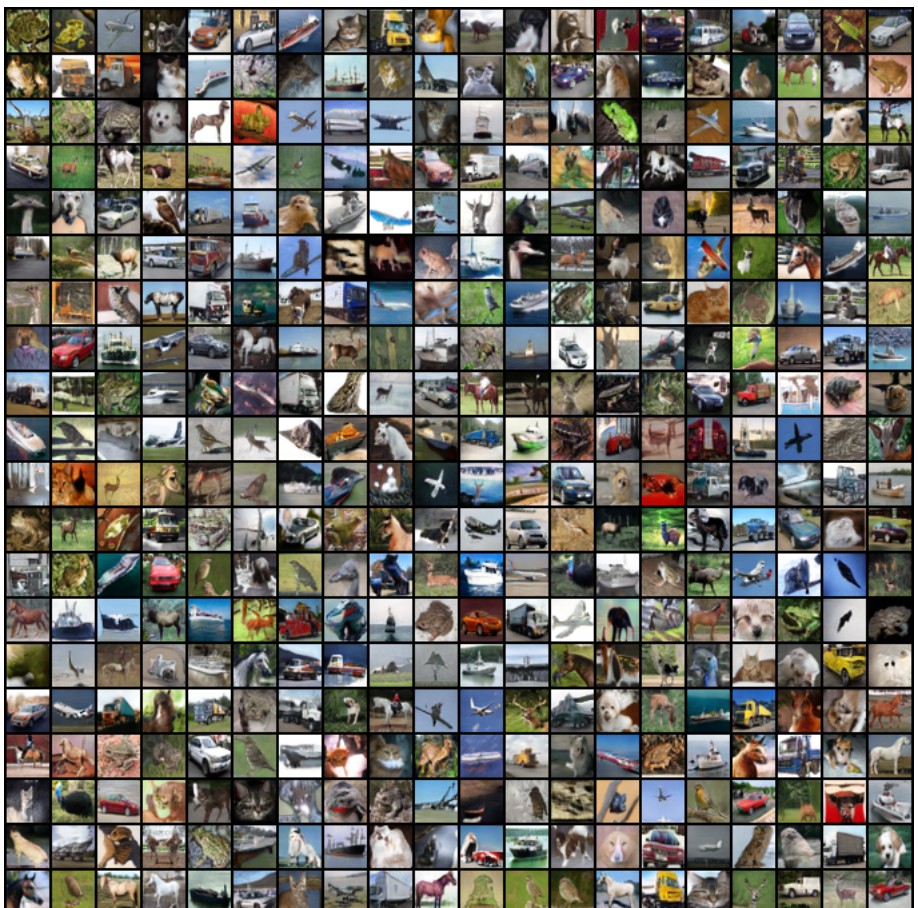

Figure 5: Generated samples from our model trained on CIFAR-10 for $\sigma = 0.1$.

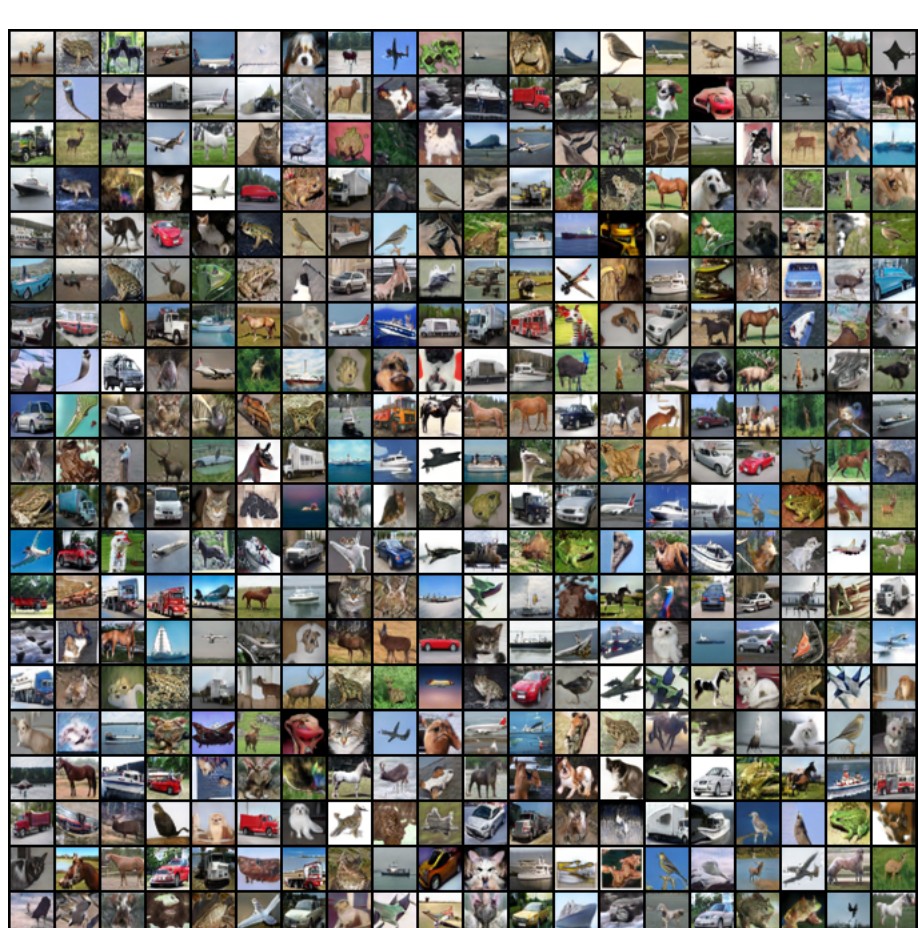

Figure 6: Generated samples from our model trained on CIFAR-10 for $\sigma = 0.5$.

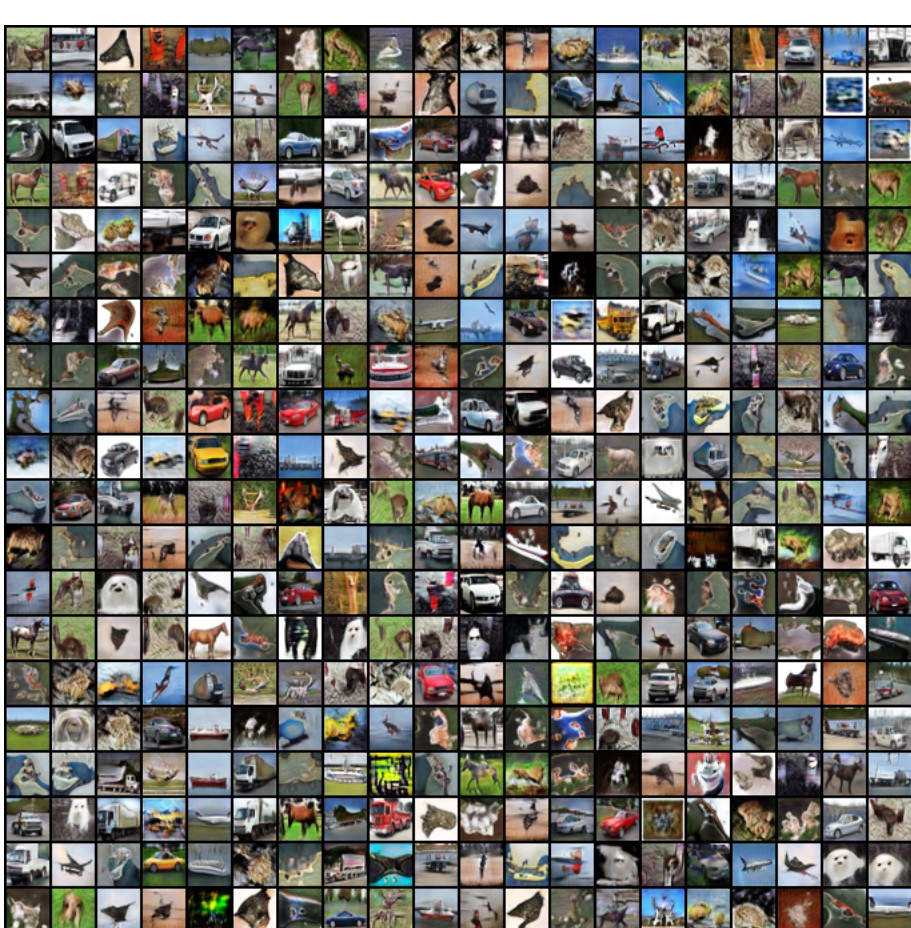

Figure 7: Generated samples from our model trained on CIFAR-10 for $\sigma = 1.0$.

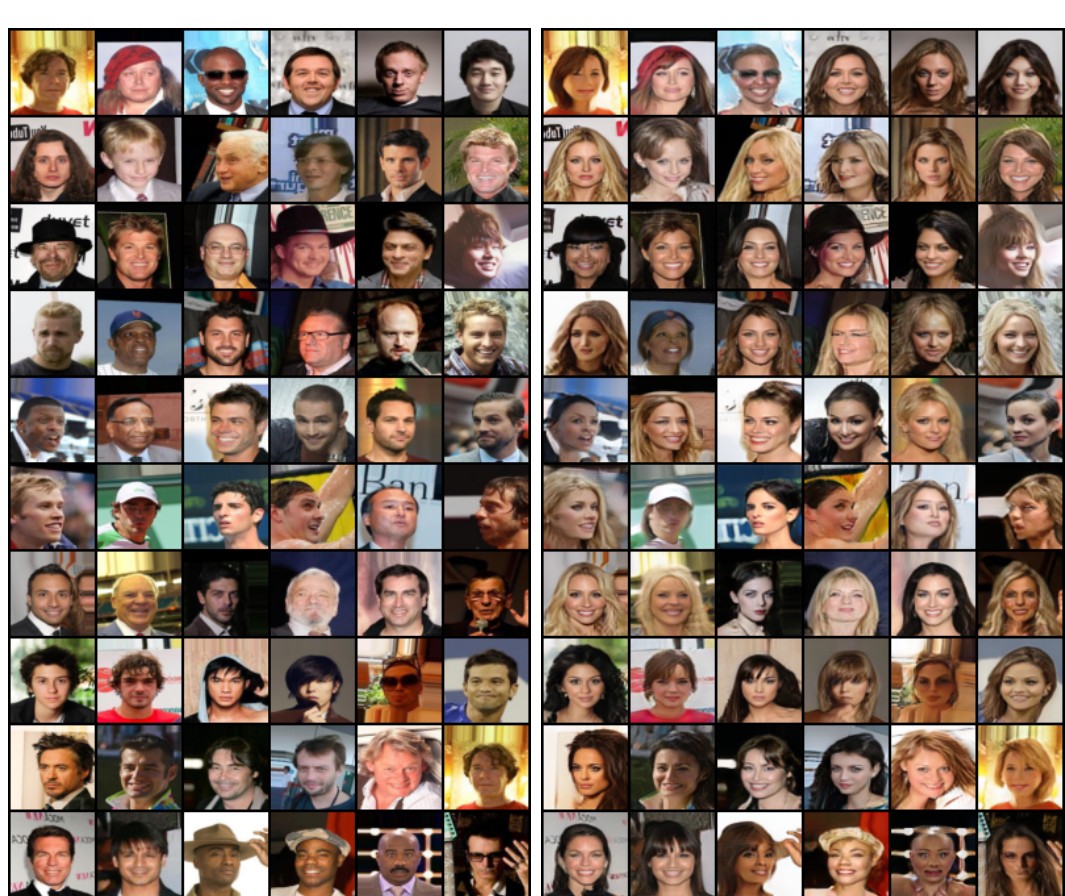

Figure 8: Unpaired $male \rightarrow female$ translation for $64 \times 64$ CelebA images.

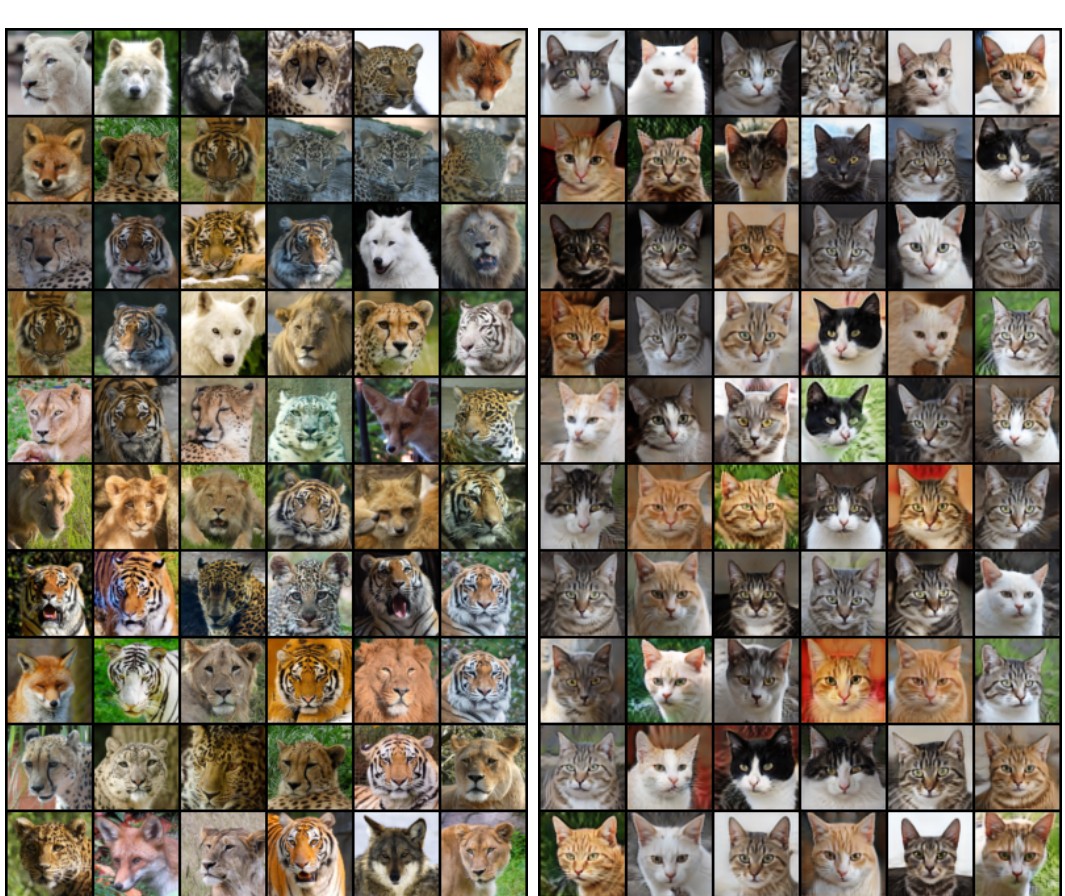

Figure 9: Unpaired $wild \rightarrow cat$ translation for $64 \times 64$ CelebA images.

