# OpenReview forum: "Scalable Simulation-free Entropic Unbalanced Optimal Transport"
_ICLR.cc/2025/Conference — Submitted to ICLR 2025_

### Official Review · Reviewer_jnr2 · 2024-11-03

**Soundness:** 3
**Presentation:** 3
**Contribution:** 3
**Rating:** 6
**Confidence:** 2

**Summary:**

This paper proposes a scalable and simulation-free approach for solving the EUOT problem. The authors derive the dynamical form and the duall formulation of the EUOT problem and their method is based on these interpretations. The paper also verifies the effectiveness of the proposed method on both toy datasets and image datasets.

**Strengths:**

The studied problem is interesting and important. The writing is mostly clear. The proposed method has good empirical performance.

**Weaknesses:**

Here are my comments:

1. How to guarantee the convergence of the proposed method?
2. What does NFE mean? The paper does not seem to clarify this term or explain why the proposed method results in a smaller NFE.
3. Since the main advantage of the proposed methods is its scability, I think the paper should provide a detailed comparison on the computational cost rather than only showing the accuracy.

**Questions:**

see weakness

---

> ### Author Response · Authors · 2024-11-22
> **Response to Reviewer jnr2**
>
> We sincerely thank the reviewer for carefully reading our manuscript and providing valuable feedback. Moreover, we appreciate the reviewer for considering our EUOT problem as "interesting and important". We hope our responses to be helpful in addressing the reviewer's concerns.
>
> $ $
>
> ---
> **W1.**
> How to guarantee the convergence of the proposed method?
>
> **A.**
> Our goal is to compute the Entropic OT transport map between continuous measures based on finite samples from each measure. Unfortunately, unlike the optimal transport problem between discrete measures, the convergence properties of adversarial approaches for learning the optimal transport map between continuous measures have never been established [1,2]. We agree with the reviewer that establishing the convergence property of our algorithm would be an important direction for future research. However, this aspect was beyond the scope of this work.
>
> $ $
>
> ---
> **W2.**
> What does NFE mean? The paper does not seem to clarify this term or explain why the proposed method results in a smaller NFE.
>
> **A.**
> We apologize for not providing a definition of NFE (Number of Function Evaluations). In the revised version of our manuscript, **we included a definition of NFE in the caption of Table 1**. NFE refers to the number of neural network inferences required to generate a single sample. In dynamical models, such as Diffusion and Flow Matching, sample generation requires a large number of NFEs ($\geq 100$).
>
> In our work, we establish the reciprocal property of the EUOT problem (Thm 3.2) and leverage this property to parametrize our model (Eq. 29). **This approach enables efficient sample generation with only 1 NFE** ($y= T_{\theta}(x,z)$ where $x$ is the input sample and $z$ is Gaussian noise for stochastic generation). Therefore, our sample generation is over $100$ times faster than that of dynamic models. Moreover, this parametrization provides efficient simulation-free training. which makes our model training over twice as fast compared to existing EOT models (Lines 424-426).
>
> $ $
>
> ---
> **W3.**
> Since the main advantage of the proposed methods is its scability, I think the paper should provide a detailed comparison on the computational cost rather than only showing the accuracy.
>
> **A.**
> Thank you for the thoughtful comment. As discussed in the response to W2, regarding computational cost, our generator parametrization using the reciprocal property (Eq. 29) allows **efficient training ($ < 50 \\%$) and sample generation ($ \approx 1\\%$) compared to existing EOT methods**.
>
> As discussed in Lines 401-405, we define the scalability challenges of existing EOT models as the difficulty of training EOT models when the distance between source and target distributions is large. In this respect, our model achieves a more accurate approximation of target distribution in both generative modeling and I2I translation tasks, without requiring any pretraining.
>
> $ $
>
> **References**
> [1] Choi, Jaemoo, Jaewoong Choi, and Myungjoo Kang. "Generative modeling through the semi-dual formulation of unbalanced optimal transport." NeurIPS 2023.
> [2] Rout, Litu, Alexander Korotin, and Evgeny Burnaev. "Generative modeling with optimal transport maps." ICLR 2022.

---

> > ### Author Response · Authors · 2024-11-25
> >
> > We sincerely thank the reviewer for the effort in reviewing our paper. We would greatly appreciate it if the reviewer let us know whether our response was helpful in addressing the reviewer's concerns. If there are additional concerns or questions, please let us know. If our responses were helpful in addressing concerns, we kindly ask the reviewer to consider raising the score.

---

> > > ### Comment · Reviewer_jnr2 · 2024-11-25
> > >
> > > Thank you for the rebuttal. Currently I will maintain my score. It seems that other reviewers have some concerns on this paper,  I'm curious about their responses to the authors' rebuttal.

---

### Official Review · Reviewer_ALVW · 2024-11-04

**Soundness:** 2
**Presentation:** 3
**Contribution:** 2
**Rating:** 5
**Confidence:** 4

**Summary:**

The authors propose a new method for solving the dynamic formulation of the Entropic Unbalanced Optimal Transport (EUOT) problem. The method is based on the Stochastic Optimal Control (SOC) formulation of the dynamic EUOT. The authors divide the optimized variables into two parts: value function of the corresponding SOC problem and the underlying distribution dynamics. Authors train them on functionals, derived from the original SOC formulation and the corresponding Hamilton-Jacobi-Bellman equation. Representation of the problem in terms of this coordinate descent results in the ability to parameterize distribution dynamics of EUOT as a stochastic one-step map instead of the stochastic process, thus allowing to achieve simulation-free (SF) procedure, which authors call SF-EUOT. Authors validate the method on toy 2d examples, unconditional generation and unpaired I2I translation, where they beat OT-based counterparts.

**Strengths:**

Originality: construction of the algorithm based on the coordinate descent with respect to the value function and the distribution dynamics, is novel. One of the optimized functionals is based on the HJB equation, which I have not seen successfully optimized in PINN-like [1] fashion in the OT context.

Quality: the paper has sound theory, well marked connections with other methods and more general OT and SOC theory.

Clarity: despite the amount of derivations, the mathematical writing is accurate and easy to follow, the experimental section and the overall presentation is clear.

Significance: the paper introduces a way to solve dynamic (essentially, multi-step) OT problems, while parameterizing the underlying distribution dynamics with a single-step map. This is especially important in case of Entropic OT problems, where one-step parameterization does not directly allow to estimate entropy, which is why many methods resort to simulation-based training.

[1] Physics-informed neural networks: A deep learning framework for solving forward and inverse problems involving nonlinear partial differential equations

**Weaknesses:**

My main concerns are related to the applicability of the method and limited comparison with the other existing models.

* First, in Table 1 the authors compare their method with models based on EOT, Diffusion and Flow Matching. While I agree with the choice of the EOT solvers, most of the chosen diffusion models correspond to results from 2021-2022. They are now outperformed by a large margin by even one-step diffusion distillation methods (CTM [1] and GDD [2] with FID around 1.6-1.7), which should be indicated for a more fair comparison. Overall, while being more efficient than other EOT solvers, the proposed SF-EUOT is still inefficient compared to more practical methods for unconditional generation. Given this, I would consider this experiment more as a benchmark for EOT solvers rather than a broad comparison with competitive methods.

* In Image-to-Image experiments and in Table 2, in particular, authors choose a limited set of methods for comparison. Two of them, DiscoGAN and CycleGAN, perform much worse than newer (but still 2-4 years old) one-step GAN-based methods such as CUT [3], StarGAN-v2 [4], and ITTR [5], which complicates fair comparison. In the Wild->Cat experiment authors choose to compare only with multi-step methods based on Schrödinger Bridge, while the results from DiscoGAN, CycleGAN and NOT, present in the Male->Female experiment, are missing. Besides, there are other SB-based methods proven competitive in practice, e.g. NSB [6], which is, being a few-step method, relevant for comparison. Overall, this section would greatly benefit from more consistent comparisons (e.g. with the same set of models for both experiments) and from adding higher-quality one-step (or few-step) baselines.

* In the Image-to-Image experiments, only FID is measured as a metric of generation quality. However, as it was done by the authors in Figure 2, it is essential to also measure some similarity metric between the input and the output. It will allow comparison of faithfulness-realism tradeoff between the proposed method and the baselines and will exclude situations e.g. in which the method achieves better FID at the cost of poor transport cost.

* Samples obtained by the SF-EUOT in the Wild->Cat translation problem (Figure 8), seem to significantly suffer from mode collapse.

[1] Consistency Trajectory Models: Learning Probability Flow ODE Trajectory of Diffusion

[2] Diffusion Models Are Innate One-Step Generators

[3] Contrastive Learning for Unpaired Image-to-Image Translation

[4] StarGAN v2: Diverse Image Synthesis for Multiple Domains

[5] ITTR: Unpaired Image-to-Image Translation with Transformers

[6] Unpaired Image-to-Image Translation via Neural Schrödinger Bridge

**Questions:**

* Could the authors clarify more on the technical implementation of the gradient norm and the Laplace operator of the value function and their computation time (compared to the standard forward and backward passes)? While it is mentioned that the Laplace operator is approximated by the Hutchinson trace estimator, here it would require calculating a Hessian-vector product, which is much more computationally expensive than the Jacobian-vector product, used in likelihood calculations.

* Did the authors test the method in the ODE scenario (with sigma = 0.0)? While it may be less interesting due to the higher feasibility of the non-entropic OT problem (e.g. solved by NOT), it should substantially decrease training time due to the lack of the Laplace operator in computations.

* Could the authors check whether there is a bug in the line 366? Here, the partial derivative with respect to time is approximated by the discrete-time approximation of the full time-derivative, which seems confusing.

* Despite training a stochastic map, in Figure 3 only one-to-one mappings are visualized. Is it a design choice, or here the generator is deterministic?

---

> ### Author Response · Authors · 2024-11-22
> **Response to Reviewer ALVW (1/2)**
>
> We sincerely thank the reviewer for carefully reading our manuscript and providing valuable feedback. Moreover, we appreciate the reviewer for considering our single-step map parametrization as "especially important in case of Entropic OT problems". We hope our responses to be helpful in addressing the reviewer's concerns. We highlighted the corresponding revisions in the manuscript in Brown.
>
> $ $
>
> ---
> **W1.** First, in Table 1 the authors compare their method with models based on EOT, Diffusion and Flow Matching. While I agree with the choice of the EOT solvers, most of the chosen diffusion models correspond to results from 2021-2022. They are now outperformed by a large margin by even one-step diffusion distillation methods (CTM [1] and GDD [2] with FID around 1.6-1.7), which should be indicated for a more fair comparison. Overall, while being more efficient than other EOT solvers, the proposed SF-EUOT is still inefficient compared to more practical methods for unconditional generation. Given this, I would consider this experiment more as a benchmark for EOT solvers rather than a broad comparison with competitive methods.
>
> **A.** We appreciate the reviewer for suggesting new SOTA distillation methods.  We included these models in our CIFAR-10 table (Table 1) under the category of Diffusion Distillation. We also agree with the reviewer that the CIFAR-10 results serve as a benchmark among EOT solvers.
> The existing Entropic OT approaches face scalability challenges. Specifically, when the distance between source and target distributions is large, the neural network for learning the Entropic Optimal Transport map fails to converge. **The main goal of our paper is to propose scalable EOT solvers that can converge on image datasets without relying on a pretraining scheme**. We would like to emphasize that our model is the only EOT models that present competitive results without relying on such a pretraining scheme.
>
> $ $
>
> ---
> **W2.** In Image-to-Image experiments and in Table 2, in particular, authors choose a limited set of methods for comparison. Two of them, DiscoGAN and CycleGAN, perform much worse than newer (but still 2-4 years old) one-step GAN-based methods such as CUT [3], StarGAN-v2 [4], and ITTR [5], which complicates fair comparison. In the Wild->Cat experiment authors choose to compare only with multi-step methods based on Schrödinger Bridge, while the results from DiscoGAN, CycleGAN and NOT, present in the Male->Female experiment, are missing. Besides, there are other SB-based methods proven competitive in practice, e.g. NSB [6], which is, being a few-step method, relevant for comparison. Overall, this section would greatly benefit from more consistent comparisons (e.g. with the same set of models for both experiments) and from adding higher-quality one-step (or few-step) baselines.
>
> **A.** We appreciate the reviewer for the valuable advice. In this paper, **we provided broad comparisons with diverse approaches on CIFAR-10 (Table 1) and focused on OT approaches in the image-to-image (I2I) translation tasks (Table 2)**. Within the OT framework, generative modeling on CIFAR-10 is more challenging than I2I translation tasks, because the source and target distributions are more distant (Gaussian $\rightarrow$ Data vs. Data1 $\rightarrow$ Data2). Therefore, we prioritized broad comparisons on CIFAR-10. In I2I tasks, each paper typically adopts different datasets and resolutions. We agree with the reviewer on the importance of establishing a unified evaluation scheme. Unfortunately, we have not yet identified such an effort in the literature. Hence, the previous EOT approaches also focused on comparisons with the most relevant baselines [1, 2]. Due to current resource constraints, we were unable to provide CUT and UNSB experimental results on our target dataset. For completeness, we will reproduce these models and incorporate the results in our manuscript.
>
>
> [1] De Bortoli, Valentin, et al. "Schr\" odinger Bridge Flow for Unpaired Data Translation." NeurIPS 2024.
> [2] Gushchin, Nikita, et al. "Adversarial Schr\" odinger Bridge Matching." NeurIPS 2024.

---

> ### Author Response · Authors · 2024-11-22
> **Response to Reviewer ALVW (2/2)**
>
> $ $
>
> ---
> **W3.** In the Image-to-Image experiments, only FID is measured as a metric of generation quality. However, as it was done by the authors in Figure 2, it is essential to also measure some similarity metric between the input and the output. It will allow comparison of faithfulness-realism tradeoff between the proposed method and the baselines and will exclude situations e.g. in which the method achieves better FID at the cost of poor transport cost.
>
> **A.**
> **We evaluated the LPIPS metric to measure the similarity between the input and the translated output**. The experimental results are provided below:
>
> - LPIPS metric ($\downarrow$)
>
> |Model| Male→Female (64x64) | Wild→Cat (64x64)|
> |:---|:---|:---|
> |DSBM| 0.246 | 0.589 |
> |Ours| **0.227** | **0.508** |
>
> These results demonstrate that our model achieves a lower transport cost (LPIPS) while also achieving better target distribution matching (FID in Table 2). We appreciate the reviewer for the constructive comments. These additional experiments provide further evidence supporting the effectiveness of our model.
>
> $ $
>
> ---
> **Q1.** Could the authors clarify more on the technical implementation of the gradient norm and the Laplace operator of the value function and their computation time (compared to the standard forward and backward passes)? While it is mentioned that the Laplace operator is approximated by the Hutchinson trace estimator, here it would require calculating a Hessian-vector product, which is much more computationally expensive than the Jacobian-vector product, used in likelihood calculations.
>
> **A.** To approximate the Laplace operator $\Delta v$, we utilize the Jacobian-vector product, not the Hessian-vector product. Specifically, we adopt the Hutchinson trace estimator as follows: (1) Sample Gaussian noise $\epsilon \sim N(0, I)$. (2) Compute the inner-product $u = \langle \epsilon, \nabla v \rangle$, where $\nabla v$ is computed by a backward pass. (3) Compute the gradient of this inner-product and calculate the inner-produce again $\langle \epsilon, \nabla u \rangle$.
>
> For the time derivative, we compared the computation time of two approaches: (1) our forward Euler-like implementation in Eq. 30 and (2) the exact derivative using a backward pass. With a batch size of 16 and two RTX 3090Ti GPUs, (1) required 0.5-0.6 second and (2) required 0.4 seconds.
>
> $ $
>
> ---
> **Q2.** Did the authors test the method in the ODE scenario (with sigma = 0.0)? While it may be less interesting due to the higher feasibility of the non-entropic OT problem (e.g. solved by NOT), it should substantially decrease training time due to the lack of the Laplace operator in computations.
>
> **A.** We conducted additional experiments with the sigma=0. As the reviewer imagined, the training time decreases to $80 \\%$. However, the performance drop was significant ($14.59 \rightarrow 19.55$ in Wild $\rightarrow$ Cat and $8.44 \rightarrow 31.9$ in Male $\rightarrow$ Female). We attribute this performance gap to the role of noise level $\sigma$. When there is an appropriate level of noise $\sigma$, our discriminator is able to observe a wider range of support for $\rho_{t}$ due to the Gaussian perturbation in Eq. 29. We believe this helps neural network training.
>
> $ $
>
> ---
> **Q3.**
> Could the authors check whether there is a bug in the line 366? Here, the partial derivative with respect to time is approximated by the discrete-time approximation of the full time-derivative, which seems confusing.
>
> **A.**
> We apply Eq. 30 to approximate the time derivative. We also tested the discrete-time derivative by replacing $x_{t + \Delta t}$ with $x_{t}$ in Eq 30. However, this performed significantly worse. We believe this happens because the value network $v_{\phi} (t + \Delta t, \cdot)$ does not observe $\rho_{t}$ during training. For clarification, we revised the term "first-order Euler discretization" in Line 365 to "the following approximation".
>
> $ $
>
> ---
> **Q4.**
> Despite training a stochastic map, in Figure 3 only one-to-one mappings are visualized. Is it a design choice, or here the generator is deterministic?
>
> **A.**
> For better clarity, we visualized one transported sample for each source sample. As explained in Lines 1107-1111, our transport map parametrization is a stochastic generator.

---

> > ### Author Response · Authors · 2024-11-25
> >
> > We sincerely thank the reviewer for the effort in reviewing our paper. We would greatly appreciate it if the reviewer let us know whether our response was helpful in addressing the reviewer's concerns. If there are additional concerns or questions, please let us know. If our responses were helpful in addressing concerns, we kindly ask the reviewer to consider raising the score.

---

> > > ### Comment · Reviewer_ALVW · 2024-11-28
> > >
> > > I thank the authors for answering my questions and addressing some of my concerns related to CIFAR-10 comparisons and measuring similarity between models' inputs and outputs.
> > >
> > > However, my main concern remains the same: while SF-EUOT shows competitive results in unconditional generation, this task is better tackled within the diffusion framework. At the same time, unpaired I2I is a problem, in which SF-EUOT, as an OT-based method, could fully demonstrate its practical value. However, this value needs to be verified by performing a deeper comparison with the baselines and by tackling the currently present issues of the model such as mode collapse (evident in the Wild->Cat experiment). Thus, I will keep my current score.

---

> ### Author Response · Authors · 2024-11-30
>
> Thank you for the follow-up questions.
>
> $ $
>
> ---
> **Q1.** However, my main concern remains the same: while SF-EUOT shows competitive results in unconditional generation, this task is better tackled within the diffusion framework. At the same time, unpaired I2I is a problem, in which SF-EUOT, as an OT-based method, could fully demonstrate its practical value. However, this value needs to be verified by performing a deeper comparison with the baselines.
>
> **A.**
> We agree with the reviewer that our evaluation of unpaired I2I could benefit from a more thorough and consistent comparison with other approaches. Following the reviewer's suggestion, **we conducted additional experiments to compare the performance of competitive methods (CUT [1] and UNSB [2]) on our I2I tasks**: Male $\rightarrow$ Female (64x64) and Wild $\rightarrow$ Cat  (64x64). The results are as follows:
>
> |Data| Male $\rightarrow$ Female | Wild $\rightarrow$ Cat|
> |---:|---:|---:|
> |CUT | 11.92 | 21.55 |
> |UNSB| 23.52 | 33.40 |
> |Ours| **8.44**  | **14.59** |
>
> **Our model outperforms both competitive methods in both tasks.** We will incorporate these results into the main text (Table 2). These additional results will emphasize the practical value of our model by providing comparisons beyond OT-based approaches. We sincerely thank the reviewer for suggesting these valuable experiments.
>
> $ $
>
> ---
> **Q.**
> This value needs to be verified by tackling the currently present issues of the model such as mode collapse (evident in the Wild->Cat experiment).
>
> **A.**
> We apologize for missing this comment in our initial response. We acknowledge the reviewer's concern about mode collapse and agree that addressing it is important. This is the common limitation across the adversarial approaches. We will add this limitation in the conclusion section. To provide further evidence, we included the **additional qualitative samples from CUT [1] and  UNSB [2] in the Wild $\rightarrow$ Cat I2I task** through the anonymous link [Anonymous link](https://github.com/Submission5695/SF-ESUOT), which is conducted in the response to Q1. Both models are adversarial approaches and exhibit more severe mode collapse problems. Additionally, **we would like to emphasize that our model achieves the best FID score in the Wild $\rightarrow$ Cat I2I task**. This result suggests that our model exhibits the least severe mode collapse problem compared to existing approaches. Nevertheless, we agree with the reviewer that addressing the mode collapse problem is an important and promising future research direction.
>
> $ $
>
> **References**
> [1] Contrastive Learning for Unpaired Image-to-Image Translation
> [2] Unpaired Image-to-Image Translation via Neural Schrödinger Bridge
>
> $ $
>
> ---
> We hope that our responses are helpful in addressing the concerns. Also, we will be happy to discuss any remaining concerns during the discussion phase. If the responses were helpful in addressing concerns, we kindly ask that the reviewer consider raising the score.

---

### Official Review · Reviewer_iXa3 · 2024-11-08

**Soundness:** 4
**Presentation:** 2
**Contribution:** 3
**Rating:** 6
**Confidence:** 4

**Summary:**

This work introduces a method to solve the (semi-relaxed) entropic unbalanced optimal transport problem. First, different equivalent formulation of the problem (namely a dynamic formulation and a dynamic dual) are derived. Then, it is shown that the problem can be stated as a stochastic optimal control problem with a HJB constraint on the value function and a Fokker-Planck constraint for the distribution. The authors then propose to solve this problem by parameterizing the value function with neural networks. Finally, they experiment on generative modeling tasks, and compare on synthetic datasets how well the method allows to recover the optimal coupling and value.

**Strengths:**

The paper is overall well written. It introduces a well motivated method by deriving a suitable dual dynamic formulation of the EUOT problem, and rewriting it as a stochastic optimal control problem.

The experiments on generative modeling seem good.

**Weaknesses:**

On the presentation of the paper, I believe some things could be improved. For instance, in Section 4.1, a lot of equations are presented, which are obtained by performing only some rewriting (e.g. equation (21) to (22) and then (22) to (24)). While this is very clear, I would suggest to just report the last equation (24) with some description of what has been done, and report the detailed derivations in Appendix. Also, the legend of Figure 3 are too small. A lot of abreviations are not given or are given after their introduction (NFE, IPF, IMF, HJB...). The notations often change between $u_t(x)$ and $u(t,x)$.

The full procedure is not very clear. It would be better to add an Algorithm.

I believe some relevant baselines are missing from the Generative modeling experiment. I am thinking e.g. of [1] who solve the UOT problem and reported a FID of 2.97 on CIFAR10.

While this method computes an unbalanced OT problem, the robustness to outliers is not discussed nor tested.

[1] Choi, Jaemoo, Jaewoong Choi, and Myungjoo Kang. "Generative modeling through the semi-dual formulation of unbalanced optimal transport." Advances in Neural Information Processing Systems 36 (2024).

**Questions:**

Line 31, the paper (Jordan et al, 1998) is cited for generative modeling. I don't think they do generative modeling at all.

Something more subjective, Line 107, I don't like the terminology of "stochastic map". It is not really a map. I would be more accurate to describe it as a (conditional) probability or probability kernel.

Since the first marginal of the problem is fixed, it would be clearer to state the problem as semi-unbalanced. Is it more difficult to derive Theorem 3.2 and Proposition 3.4 with both marginals relaxed?

The method relaxes a lot of constraints into the loss. I wonder how hard it is to make sure every term go to 0. Also, is this method costly compared e.g. to the semi-dual approach of [1]? And is it stable to train?

Line 512, it is stated that "there is no closed-form for the EUOT problem". But I believe there are in [2], and [2] is also the right reference for closed forms of the EOT problem between Gaussian. Therefore, I would suggest to also perform the experiments in Section 5.2 with unbalanced Gaussian.

Some references on generative modeling with unbalanced OT problem could be added, e.g. [3, 4].

Typos:
- Line 175: "Dynamcial"
- Line 513: "Conducted same benchmark"

[1] Choi, Jaemoo, Jaewoong Choi, and Myungjoo Kang. "Generative modeling through the semi-dual formulation of unbalanced optimal transport." Advances in Neural Information Processing Systems 36 (2024).

[2] Janati, H., Muzellec, B., Peyré, G., & Cuturi, M. (2020). Entropic optimal transport between unbalanced gaussian measures has a closed form. Advances in neural information processing systems, 33, 10468-10479.

[3] Dao, Q., Ta, B., Pham, T., & Tran, A. (2023). Robust Diffusion GAN using Semi-Unbalanced Optimal Transport. arXiv preprint arXiv:2311.17101.

[4] Lübeck, F., Bunne, C., Gut, G., del Castillo, J. S., Pelkmans, L., & Alvarez-Melis, D. (2022). Neural unbalanced optimal transport via cycle-consistent semi-couplings. arXiv preprint arXiv:2209.15621.

---

> ### Author Response · Authors · 2024-11-22
> **Response to Reviewer iXa3 (1/2)**
>
> We appreciate the reviewer for carefully reading our manuscript and providing valuable feedback. We hope our responses adequately address the reviewer's concerns. We highlighted the corresponding revisions in the manuscript in Blue.
>
> $ $
>
> ---
> **W1.** On the presentation of the paper, I believe some things could be improved. For instance, in Section 4.1, a lot of equations are presented, which are obtained by performing only some rewriting (e.g. equation (21) to (22) and then (22) to (24)). While this is very clear, I would suggest to just report the last equation (24) with some description of what has been done, and report the detailed derivations in Appendix. Also, the legend of Figure 3 are too small. A lot of abreviations are not given or are given after their introduction (NFE, IPF, IMF, HJB...). The notations often change between $u_{t}(x)$ and $u(t,x)$.
>
> **A.**
> Thank you for the thoughtful comment. Following the reviewer's advice, we revised our manuscript as follows:
> - We omitted Eq. 21 from the original manuscript, because we considered the calculation from Eq.17 to Eq.22 to be relatively straightforward. We were unable to move Eq. [22-23] to the Appendix, because this equation is referenced for Eq. 25.
> - We revised the legend of Fig. 3.
> - We added the definitions of the abbreviations (NFE, HJB) in the manuscript. In our paper, IPF [1] and IMF [2] refer to the previous methods.
> - We unified the notation for $u_{t}(x)$ and $u(t,x)$ to $u_{t}(x)$.
>
> $ $
>
> ---
> **W2.** The full procedure is not very clear. It would be better to add an Algorithm.
>
> **A.** In the original version of our manuscript, we included Algorithm 1 on Page 5.
>
> $ $
>
> ---
> **W3.** I believe some relevant baselines are missing from the Generative modeling experiment. I am thinking e.g. of [3] who solve the UOT problem and reported a FID of 2.97 on CIFAR10.
>
> **A.**
> Thank you for suggesting the relevant reference. We included UOTM [3] (FID = 2.97) and OTM [4] (FID = 7.68) in our CIFAR-10 result table (Table 1) under the class of OT.
>
> $ $
>
> ---
> **W4.** While this method computes an unbalanced OT problem, the robustness to outliers is not discussed nor tested.
>
> **A.** The existing Entropic OT approaches face scalability challenges. Specifically, when the distance between source and target distributions is large, the neural network for learning the Entropic Optimal Transport map fails to converge. We consider **this scalability challenge to be a critical problem in Entropic OT approaches, because this challenge undermines a key advantage of Entropic OT approaches over diffusion models**, i.e., generalizability. In this work, to address this scalability challenge, we focused on the improved optimization stability of the unbalanced OT problem [5]. Nevertheless, we agree with the reviewer that investigating the outlier robustness of the EUOT problem is a promising direction for future research.
>
> $ $
>
> ---
> **Q1.** Line 31, the paper (Jordan et al, 1998) is cited for generative modeling. I don't think they do generative modeling at all.
>
> **A.** Thank you for pointing this out. We removed this citation from Line 31.
>
> $ $
>
> ---
> **Q2.** Something more subjective, Line 107, I don't like the terminology of "stochastic map". It is not really a map. I would be more accurate to describe it as a (conditional) probability or probability kernel.
>
> **A.**
> Thank you for the suggestion. We revised Line 107 as follows:
> > Moreover, when $\sigma >0$, the optimal transport map is stochastic, i.e. $\pi^\star(\cdot|x)$ is a conditional probability distribution.
>
> $ $
>
> ---
> **Q3.** Since the first marginal of the problem is fixed, it would be clearer to state the problem as semi-unbalanced. Is it more difficult to derive Theorem 3.2 and Proposition 3.4 with both marginals relaxed?
>
> **A.**
> Thank you for the insightful comment. As the reviewer asked, the general unbalanced optimal transport problem relaxes both marginals in the source and the target. However, in our case, the transport dynamics $\rho_{t} = \text{Law}(X_{t}^{u})$ are governed by the following equation when deriving the dual formulations: $$ dX^u_t = u(t,X^u_t) dt + \sigma dW_t, \quad \ X^u_0 \sim \mu. $$ Hence, since our dynamics start with sampling from the source distribution $\mu$, we cannot relax the marginals in the source. We need to fix the starting distributions $\rho_{0} = \mu$.
>
> $ $

---

> ### Author Response · Authors · 2024-11-22
> **Response to Reviewer iXa3 (2/2)**
>
> ---
> **Q4.** The method relaxes a lot of constraints into the loss. I wonder how hard it is to make sure every term go to 0. Also, is this method costly compared e.g. to the semi-dual approach of [3]? And is it stable to train?
>
> **A.** For example, when training our model on CIFAR-10, the generator loss and value loss in Lines 8-9 of the Algorithm exhibit reasonably small absolute values about $\sim O(10^{-1})$. In practice, it is not feasible to drive each loss to exactly 0. This might be a reason why our model exhibits comparable or slightly worse numerical accuracy in Table 3. Because we are adopting a PINN-style loss function (HJB condition term in Eq. 26), if the loss does not converge to exactly zero, the value function does not precisely satisfy the optimality condition.
>
> Regarding training cost, as discussed in Lines 424-426, our model is more than twice as fast compared to other EOT models. When compared to the unbalanced optimal transport map without entropic regularization [3], our model is approximately twice as slow. This is due to the entropic regularizer, which requires the Hutchinson-Skilling trace estimator to estimate the laplacian $\triangle v_{\phi}$ in Line 7 of the Algorithm.
>
> $ $
>
> ---
> **Q5.** Line 512, it is stated that "there is no closed-form for the EUOT problem". But I believe there are in [2], and [2] is also the right reference for closed forms of the EOT problem between Gaussian. Therefore, I would suggest to also perform the experiments in Section 5.2 with unbalanced Gaussian.
>
> **A.** [2] provides the closed-form solution for the EUOT problem when both marginals are fixed. However, as we discussed in Q3, our goal is to solve the EUOT problem when the source distribution is fixed. Therefore, the solution in [2] is not applicable to our target problem. Consequently, we evaluated the numerical accuracy of our model in the EOT problem setting, as shown in Table 3.
>
> $ $
>
> ---
> **Q6.** Some references on generative modeling with unbalanced OT problem could be added, e.g. [7, 8].
>
> **A.** Thank you for suggesting these valuable references. We included [7,8] in the Introduction section.
>
> $ $
>
> ---
> **Typos:**
>
> **A.** Thank you for the careful advice. We corrected the manuscript accordingly.
>
> $ $
>
> **References**
>
> [1] Chen, Tianrong, Guan-Horng Liu, and Evangelos A. Theodorou. "Likelihood training of schr\" odinger bridge using forward-backward sdes theory." ICLR 2022.
> [2] Shi, Yuyang, et al. "Diffusion Schrödinger bridge matching." NeurIPS 2023.
> [3] Choi, Jaemoo, Jaewoong Choi, and Myungjoo Kang. "Generative modeling through the semi-dual formulation of unbalanced optimal transport." NeurIPS 2023.
> [4] Rout, Litu, Alexander Korotin, and Evgeny Burnaev. "Generative modeling with optimal transport maps." ICLR 2022.
> [5] Choi, Jaemoo, Jaewoong Choi, and Myungjoo Kang. "Analyzing and Improving Optimal-Transport-based Adversarial Networks." ICLR 2024.
> [6] Janati, Hicham, et al. "Entropic optimal transport between unbalanced gaussian measures has a closed form." NeurIPS 2020.
> [7] Dao, Quan, et al. "Robust Diffusion GAN using Semi-Unbalanced Optimal Transport." Arxiv.
> [8] Lübeck, Frederike, et al. "Neural unbalanced optimal transport via cycle-consistent semi-couplings." Arxiv.

---

> > ### Comment · Reviewer_iXa3 · 2024-11-23
> >
> > Thank you for your rebuttal and revising the paper. I have still some comments.
> >
> > **In the original version of our manuscript, we included Algorithm 1 on Page 5.** Sorry, I somehow missed it when writing the review.
> >
> > **Nevertheless, we agree with the reviewer that investigating the outlier robustness of the EUOT problem is a promising direction for future research.** I think it would be fairly easy to add some outliers to a 2D distribution (e.g. Gaussian), and assess that it is robust to outliers. For instance in [1], they do it on a 1D mixture. I would recommend to do this as it is probabibly the main feature of why we would wand to use Unbalanced Optimal Transport, at least in my opinion.
> >
> > **since our dynamics start with sampling from the source distribution $\mu$, we cannot relax the marginals in the source.** Since the method can only compute semi-unbalanced entropic OT, I would recommend to make it more clear in the paper, and in particular to change the title accordingly, as for now, it is a bit misleading.
> >
> > **[2] provides the closed-form solution for the EUOT problem when both marginals are fixed. However, as we discussed in Q3, our goal is to solve the EUOT problem when the source distribution is fixed.** I am not sure to understand. In the paper, you use the closed-form of EOT in the Gaussian-to-Gaussian case as done in [3]. But, from my understanding, in [3], they provide the closed form of the Schrödinger bridge between Gaussian, while in [2], they provided the closed form for the EOT and EUOT problems between Gaussian. Maybe it is possible to derive the semi-unbalanced case from their results.
> >
> > Nonetheless, it is stated line 508 "First, the EOT problem between Gaussians has a closed-form solution for the optimal transport coupling, while there is no such closed form solution for the EUOT problem", which is not true, and I believe you still should add [2] in addition to [3] as a reference.
> >
> >
> >
> > [1] Choi, J., Choi, J., & Kang, M. (2024). Generative modeling through the semi-dual formulation of unbalanced optimal transport. Advances in Neural Information Processing Systems, 36.
> >
> > [2] Janati, H., Muzellec, B., Peyré, G., & Cuturi, M. (2020). Entropic optimal transport between unbalanced gaussian measures has a closed form. Advances in neural information processing systems, 33, 10468-10479.
> >
> > [3] Bunne, C., Hsieh, Y. P., Cuturi, M., & Krause, A. (2023, April). The schrödinger bridge between gaussian measures has a closed form. In International Conference on Artificial Intelligence and Statistics (pp. 5802-5833). PMLR.

---

> ### Author Response · Authors · 2024-11-25
>
> Thank you for the follow-up. We are preparing for the response. We are working on the outlier experiments and also carefully investigating whether we can obtain the closed-form solution for the Entropic Semi-Unbalanced Optimal Transport between Gaussian distributions.

---

> ### Author Response · Authors · 2024-11-29
>
> Thank you for the follow-up questions.
>
> $ $
>
> ---
> **Q.** Nevertheless, we agree with the reviewer that investigating the outlier robustness of the EUOT problem is a promising direction for future research. I think it would be fairly easy to add some outliers to a 2D distribution (e.g. Gaussian), and assess that it is robust to outliers. For instance in [1], they do it on a 1D mixture. I would recommend to do this as it is probabibly the main feature of why we would wand to use Unbalanced Optimal Transport, at least in my opinion.
>
> **A.** We agree with the reviewer that the outlier robustness is the main feature of using the Unbalanced Optimal Transport. **Following the reviewer's suggestion, we conducted additional experiments to demonstrate the outlier robustness of the Entropic Semi-Unbalanced Optimal Transport (ESUOT) problem as in [1].** These results are included in Appendix D.1 and Fig 4. Due to page constraints, we were unable to include this result in our main text. Hence, we revised our main text to highlight this outlier robustness in Line 368.
>
> **We tested our SF-ESUOT model and the SF-EOT model (our model for the EOT problem without unbalancedness) using a synthetic dataset containing $2\\%$ outliers.** The results are similar to those in [1]. The SF-EOT model tries to match both in-distribution and outlier densities. However, this leads to the undesirable behavior of generating density outside the target support (around $x=-1$). In contrast, the SF-ESUOT problem focuses on matching the in-distribution density, demonstrating outlier robustness. These additional results will emphasize the motivation for adopting the ESUOT problem.  We appreciate the reviewer for suggesting these valuable experiments.
>
> $ $
>
> [1] Choi, Jaemoo, Jaewoong Choi, and Myungjoo Kang. "Generative modeling through the semi-dual formulation of unbalanced optimal transport." NeurIPS 2023.
>
> $ $
>
> ---
> **Q.** Since our dynamics start with sampling from the source distribution, we cannot relax the marginals in the source. Since the method can only compute semi-unbalanced entropic OT, I would recommend to make it more clear in the paper, and in particular to change the title accordingly, as for now, it is a bit misleading.
>
> **A.**
> Thank you for providing valuable advice to enhance the clarity of this work. **We revised the title, abstract, model name, and the entire manuscript to clarify that our goal is to solve the Entropic Semi-Unbalanced Optimal Transport problem.**
>
> $ $
>
> ---
> **Q.**
> [2] provides the closed-form solution for the EUOT problem when both marginals are fixed. However, as we discussed in Q3, our goal is to solve the EUOT problem when the source distribution is fixed. I am not sure to understand. In the paper, you use the closed-form of EOT in the Gaussian-to-Gaussian case as done in [3]. But, from my understanding, in [3], they provide the closed form of the Schrödinger bridge between Gaussian, while in [2], they provided the closed form for the EOT and EUOT problems between Gaussian. Maybe it is possible to derive the semi-unbalanced case from their results.
>
> Nonetheless, it is stated line 508 "First, the EOT problem between Gaussians has a closed-form solution for the optimal transport coupling, while there is no such closed form solution for the EUOT problem", which is not true, and I believe you still should add [2] in addition to [3] as a reference.
>
>  $ $
>
> **A.** We appreciate the reviewer for suggesting additional references regarding the closed-form solution for the ESUOT problem. [3] address the Schrödinger bridge problem (dynamic EOT problem), rather than the ESUOT problem. Also, there are some technical challenges for directly generalizing the closed-form solution in [2], where both marginals are relaxed, e.g. investigating the well-definedness of the Sinkhorn transform or designing optimal potential functions. Unfortunately, we were unable to derive the closed-form solution for the ESUOT problem. However, **we will revise our manuscript to additionally cite [2] as follows:**
>
> > First, the EOT problem between Gaussians has a closed-form solution for the optimal transport coupling [3], while there is no such closed-form solution for the EUOT problem. Note that [2] derived the EUOT solution for Gaussian distributions where both marginal distributions are relaxed. However, this closed-form solution is not directly applicable to our ESUOT problem.
>
> $ $
> [2] Janati, Hicham, et al. "Entropic optimal transport between unbalanced gaussian measures has a closed form." NeurIPS 2020.
> [3] Bunne, Charlotte, et al. "The schrödinger bridge between gaussian measures has a closed form." AISTATS 2023.
>
> $ $
>
> ---
> We hope that our responses are helpful in addressing the concerns. Also, we will be happy to discuss any remaining concerns during the discussion phase. If the responses were helpful in addressing concerns, we kindly ask that the reviewer consider raising the score.

---

> > ### Comment · Reviewer_iXa3 · 2024-11-29
> >
> > Thank you for addressing my last concerns. I will raise my score to 6.

---

> > > ### Author Response · Authors · 2024-11-30
> > >
> > > Thank you for reviewing our paper and providing valuable feedback. The insightful feedback from the reviewer has been a significant help in improving the quality of our manuscript.

---

### Official Review · Reviewer_yGvj · 2024-11-09

**Soundness:** 2
**Presentation:** 3
**Contribution:** 2
**Rating:** 5
**Confidence:** 3

**Summary:**

This paper proposes a generalization of the Schrödinger Bridge Problem called Entropic Unbalanced Optimal Transport (EUOT). Leveraging a stochastic optimal control interpretation, the authors efficiently solve the dual formulation of EUOT. Through convex optimization, they optimize the value $V$ and path measure $\rho_t$ separately, resulting in two loss functions to construct a fully simulation-free algorithm. The paper claims that EUOT achieves one-step generation and improves generative model scalability, although the experiments don’t clearly support this level of performance.

**Strengths:**

1. The work demonstrates novelty as the first EOT model that does not require pretraining. Based on the known equivalence between Schrödinger Bridge (SB) and EOT, the authors introduced EUOT by fixing the source distribution and proved that it generalizes EOT (and thus SB).

2. Thanks to its reciprocal property and static generator, EUOT is a simulation-free model that does not require SDE or ODE simulations for sampling.

**Weaknesses:**

1. Theoretically, compared to EOT, EUOT only adds an $f$-divergence penalty term in the objective to relax the marginal constraint, which doesn’t seem like a groundbreaking innovation.

2. While the results surpass other OT methods, EUOT does not achieve the best FID scores compared to other image generation models (Tab. 1). Additionally, Fig. 3 indicates that EUOT does not fully approximate a precise optimal transport plan, and Tab. 3 shows that EUOT performs slightly worse than some benchmarks.

**Questions:**

1. In Eq. (7), it appears that a $1 \over \sigma^2$ term may be missing in the objective. Could the authors provide a detailed derivation of Eq. (7) or reference a source that confirms this formulation? This would help clarify whether the equation is correct as presented or if any term is indeed missing.
2. The paper lacks a visualization of wild-to-cat transformations, which would help assess visual quality and compare with DSBM results at 512x512 resolution. Could the authors include wild-to-cat visualizations and provide a more detailed comparison with DSBM? Additionally, an explanation for the high FID in Table 2 compared to DSBM’s performance would clarify any discrepancies in resolution or model effectiveness.
3. Line 458 claims that Table 2 demonstrates the model's superior scalability; however, I don’t observe any changes in model size, and the dataset resolution remains the same. Could the authors clarify this claim?

---

> ### Author Response · Authors · 2024-11-22
> **Response to Reviewer yGvj (1/2)**
>
> We appreciate the reviewer for carefully reading our manuscript and providing valuable feedback. Moreover, we are grateful for the reviewer's acknowledgment that "our model is the first EOT model that does not require pretraining and is simulation-free". We hope our responses adequately address the reviewer's concerns. We highlighted the corresponding revisions in the manuscript in Red.
>
> $ $
>
> ---
> **W1.** Theoretically, compared to EOT, EUOT only adds an $f$-divergence penalty term in the objective to relax the marginal constraint, which doesn’t seem like a groundbreaking innovation.
>
> **A.** We would like to emphasize that, while the difference between EOT and EUOT lies in the additional $f$-divegence penalty term to relax the marginal constraint, **their behavior and theoretical properties are significantly different**. The unbalanced version of (Entropic) Optimal Transport (OT) offers various advantages, such as improved stability in neural network optimization [1] and robustness to outliers [2,3]. This additional stability is also observed in our CIFAR-10 experiments (Table 1), where our EUOT-Soft model achieves a superior FID score of 3.02, compared to the FID score of 4.05 achieved by our EOT model.
>
> Additionally, we would like to highlight that **our work represents the first E(U)OT approach that enables simulation-free training and efficient NFE-1 generation**. This is possible by proving that the EUOT solution satisfies the reciprocal property (Thm 3.2) and by introducing a novel parametrization based on this property in Eq. 29.
>
> $ $
>
> ---
> **W2.** While the results surpass other OT methods, EUOT does not achieve the best FID scores compared to other image generation models (Tab. 1). Additionally, Fig. 3 indicates that EUOT does not fully approximate a precise optimal transport plan, and Tab. 3 shows that EUOT performs slightly worse than some benchmarks.
>
> **A.** We agree that our SF-EUOT model does not achieve the best FID score compared to all image generation models. However, we would like to emphasize that our model is only outperformed by diffusion models, which require a significantly higher number of NFEs ($\geq 100$). **Considering that our model provides efficient generation with only 1 NFE, our model achieves competitive results.** Especially, our model outperforms all other Entropic OT (Schr  ̈odinger bridge) models, which rely on pretraining with diffusion model training scheme and require a large number of NFEs ($\geq 100$).
>
> Among the Entropic OT models, it is particularly important that **our model achieves the best results without relying on pretraining via a diffusion model training scheme**. A key advantage of Entropic OT models over diffusion models is their generalizability; they can be applied to transport arbitrary distributions, such as Image-to-Image translation tasks, since they do not depend on a manually designed forward process. However, **if Entropic OT models require pretraining using diffusion models, they remain constrained to generative modeling applications**. Consequently, our approach addresses this limitation. Furthermore, our model outperforms previous OT models [4, 5, 6] on the Image-to-Image translation task (Table 2). These results demonstrate better performance of our model in general distribution transport problems.
>
> In the synthetic dataset experiments (Table 3 and Fig 3), we acknowledge that our model exhibits comparable or slightly worse numerical accuracy compared to existing methods.  As discussed in the limitation part of the Conclusion, we hypothesize that this is due to the inherent difficulty of achieving precise matching using a PINN-style loss function. We believe that exploring alternative approaches to replace PINN-like loss functions would be a promising future research. Nevertheless, we consider that the improved scalability of our model, i.e., its ability to converge without pretraining and its superior performance on high-dimensional image datasets, can be a meaningful contribution to the Entropic OT models.
>
> $ $
>
> ---
> **Q1.** In Eq. (7), it appears that a $\frac{1}{\sigma^{2}}$ term may be missing in the objective. Could the authors provide a detailed derivation of Eq. (7) or reference a source that confirms this formulation? This would help clarify whether the equation is correct as presented or if any term is indeed missing.
>
> **A.** We omitted $\frac{1}{\sigma^{2}}$ because $\sigma$ is assumed to be constant (Eq. 3). Therefore, the minimization problem in Eq. 7 remains equivalent regardless of the inclusion of $\frac{1}{\sigma^{2}}$. To avoid confusion, we included $\frac{1}{\sigma^{2}}$ to Eq.7 as follows:
> $$
>     \inf_u \left[ \frac{1}{\sigma^2}  \int_0^1 \int_{\mathcal{X}} \frac{1}{2} \\| u_t(x) \\|^2 d\rho_t (x) dt \right], \\,\\,
>         {\rm s.t.} \\,\\, \partial_t \rho_t + \nabla \cdot (u_t \rho_t ) - \frac{\sigma^2}{2} \Delta \rho_t = 0, \\,\\, \rho_0 = \mu, \rho_1 = \nu.
> $$

---

> > ### Author Response · Authors · 2024-11-22
> > **Response to Reviewer yGvj (2/2)**
> >
> > ---
> > **Q2.** The paper lacks a visualization of wild-to-cat transformations, which would help assess visual quality and compare with DSBM results at 512x512 resolution. Could the authors include wild-to-cat visualizations and provide a more detailed comparison with DSBM? Additionally, an explanation for the high FID in Table 2 compared to DSBM’s performance would clarify any discrepancies in resolution or model effectiveness.
> >
> > **A.** As explained in Line 460, we included the wild-to-cat translation results in Figure 8 in Appendix D. Also, in Table 2, the reported FID scores for DSBM and [6] are taken from [6]. To clarify, we will revise the caption of Table 2 as follows:
> >
> > > FID score for Image-to-image Translation Tasks. All FID scores of other models are taken from their original paper, except for DSBM, which is taken from [6] and [7].
> >
> > $ $
> >
> > ---
> > **Q3.** Line 458 claims that Table 2 demonstrates the model's superior scalability; however, I don’t observe any changes in model size, and the dataset resolution remains the same. Could the authors clarify this claim?
> >
> > **A.**
> > Thank you for the thoughtful comment. In Line 458, the "superior scalability" of our model refers to **its ability to better approximate the target distribution, as measured by FID scores, compared to existing OT and EOT models**. To clarify, we revised our manuscript as follows:
> >
> > > These results demonstrate that our model achieves a better approximation of the target distributions in I2I translation tasks, compared to other OT-based approaches.
> >
> > As discussed in Lines 401-405, we define the scalability challenges of existing EOT models as the difficulty of training EOT models when the distance between source and target distributions is large. To address this challenge in generative modeling, existing dynamic Entropic OT models exploit a pretraining scheme based on the diffusion model training strategy. However, **our model achieves a more accurate approximation of target distribution in both generative modeling and I2I translation tasks, without any pretraining**. Note that in I2I translation tasks, existing dynamic Entropic OT models cannot utilize a pretraining scheme, since we cannot manually design an appropriate forward process.
> >
> > $ $
> >
> > **References**
> >
> > [1] Choi, Jaemoo, Jaewoong Choi, and Myungjoo Kang. "Analyzing and Improving Optimal-Transport-based Adversarial Networks." ICLR 2024.
> > [2] Choi, Jaemoo, Jaewoong Choi, and Myungjoo Kang. "Generative modeling through the semi-dual formulation of unbalanced optimal transport." NeurIPS 2023.
> > [3] Gazdieva, Milena, et al. "Light Unbalanced Optimal Transport." NeurIPS 2024.
> > [4] Korotin, Alexander, Daniil Selikhanovych, and Evgeny Burnaev. "Neural optimal transport." ICLR 2023.
> > [5] Shi, Yuyang, et al. "Diffusion Schrödinger bridge matching." NeurIPS 2023.
> > [6] De Bortoli, Valentin, et al. "Schr\" odinger Bridge Flow for Unpaired Data Translation.", NeurIPS 2024.
> > [7] Gushchin, Nikita, et al. "Adversarial Schr\" odinger Bridge Matching.", NeurIPS 2024.

---

> > > ### Author Response · Authors · 2024-11-25
> > >
> > > We sincerely thank the reviewer for the effort in reviewing our paper. We would greatly appreciate it if the reviewer let us know whether our response was helpful in addressing the reviewer's concerns. If there are additional concerns or questions, please let us know. If our responses were helpful in addressing concerns, we kindly ask the reviewer to consider raising the score.

---

> > > > ### Comment · Reviewer_yGvj · 2024-11-26
> > > >
> > > > Thank you for the responses and for addressing my questions. The authors have clarified my concerns in the paper. While I appreciate their efforts to achieve "simulation-free" methods, I believe the trade-off in performance limits its overall contribution. Nevertheless, I would like to raise my score to 5.

---

> > > > > ### Author Response · Authors · 2024-11-30
> > > > >
> > > > > Thank you for reviewing our paper and providing valuable comments. We are pleased that we could address the reviewer’s concerns. Also, we appreciate the reviewer for acknowledging our efforts to develop simulation-free approaches.

---

### Official Review · Reviewer_RWwA · 2024-11-11

**Soundness:** 3
**Presentation:** 2
**Contribution:** 3
**Rating:** 6
**Confidence:** 4

**Summary:**

The authors establish several new connections between entropic unbalanced optimal transport (EUOT), the Schrödinger Bridge (SB) problem, stochastic control (SOC), and action matching. Furthermore, based on these new connections, the authors derive the dual formulation of the EUOT problem and its optimality conditions. In turn, this novel dual formulation allowed authors to propose a novel maximin simulation-free solver for the EUOT problem. The authors evaluate the proposed solver on 2D toy experiments, experiments with Gaussians, image generation (CIFAR-10), and unpaired image translation (Celeba Male to Female and Wild to Cat datasets).

**Strengths:**

1) The novel dual formulation of Entropic Unbalanced Optimal Transport based on the connections between Entropic Optimal Transport, Schrödinger Bridge, and Action Matching seems promising in unifying the mentioned problems.

2) The obtained experimental results support the author's claim that the new simulation-free solver outperforms the previous approach for SB/Entropic Optimal Transport.

3) The results for image generations presented for CIFAR-10 are comparable with the diffusion/flow models while requiring only 1 NFE for inference.

**Weaknesses:**

1) While the general theory is clear and strict, some questions arise about the claims in sections 4.1 and 4.2 regarding the description and justification of the proposed method (see questions).

**Questions:**

1) In lines 308-309, the authors propose considering the dual formulation of Eq. 23 given by Eq. 25. However, Eq. 23 depends only on $V_1$, i.e., only the values of $V$ in the terminal time $t=1$, while Eq. 25 depends on the all-time moments of $V_t$. Hence, the optimization result given by Eq. 25 should depend only on the values of $V$ at the terminal times, which seems strange. Could you comment on this?

2) Does Eq. 25 require $V$ to be optimal, i.e., satisfy the HJB equation, or can it be used for any time-dependent function $V$? It is interesting because you propose to use the regularizer for the HJB condition in Eq. 26. In practice, at the arbitrary step of the algorithm, the HJB condition will unlikely hold so that it would influence the optimization of $\rho$.

3) In the algorithm (lines 229-230), you propose simultaneously performing optimization updates for both generator $T_{\theta}$ and value network $V$. However, the motivation for this choice is not clear to me. In Section 4.1, you say that you need to solve an additional optimization problem (Eq. 25) for $\rho$, which makes me think that this problem (Eq. 25) should be solved before updating the value function $V$.

---

> ### Author Response · Authors · 2024-11-22
> **Response to Reviewer RWwA**
>
> We sincerely thank the reviewer for carefully reading our manuscript and providing valuable feedback. Moreover, we appreciate the reviewer for considering our dual formulation of the EUOT problem as "promising in unifying the EOT, SB, and Action Matching problems". We hope our responses to be helpful in addressing the reviewer's concerns.
>
> $ $
>
> ---
> **Q1.** In lines 308-309, the authors propose considering the dual formulation of Eq. 23 given by Eq. 25. However, Eq. 23 depends only on $V_1$, i.e., only the values of $V$ in the terminal time $t=1$, while Eq. 25 depends on the all-time moments of $V_t$. Hence, the optimization result given by Eq. 25 should depend only on the values of $V$ at the terminal times, which seems strange. Could you comment on this?
>
> **A.** Eq. 23 is a minimization problem with respec to $(u, \rho)$. As explained in Line 994 in Appendix A.2, the optimization problem in Eq.25 is derived by establishing the relationship between the optimal $u^{\star}$ and the value function $V(t,x)$ in Eq 19. Consequently, note that Eq. 25 becomes a minimization problem solely with respect to $\rho$.
>
> $ $
>
> ---
> **Q2.** Does Eq. 25 require $V$ to be optimal, i.e., satisfy the HJB equation, or can it be used for any time-dependent function $V$? It is interesting because you propose to use the regularizer for the HJB condition in Eq. 26. In practice, at the arbitrary step of the algorithm, the HJB condition will unlikely hold so that it would influence the optimization of $\rho$.
>
> **A.**
> As the reviewer asked, during training, $V$ does not satisfy the optimality condition in Eq 25. Intuitively, we understand $V$ in Eq 25 plays a similar role as the discriminator in GAN. Specifically, the $V$-dependent term serves as a weight for each sample from $\rho_{t}$.
>
> $ $
>
> ---
> **Q3.**
> In the algorithm (lines 229-230), you propose simultaneously performing optimization updates for both generator $T_\theta$ and value network $V$. However, the motivation for this choice is not clear to me. In Section 4.1, you say that you need to solve an additional optimization problem (Eq. 25) for $\rho$, which makes me think that this problem (Eq. 25) should be solved before updating the value function $V$.
>
> **A.**
> Thank you for the careful comment. As the reviewer commented, the optimization problem for $\rho$ in Eq.25 can be interpreted as the inner-loop optimization problem in Eq. 24. In practice, we adopt an adversarial optimization algorithm by alternatively performing gradient descent on the value function $V$ (Eq. 27) and the generator network $T$ (Eq. 28). Therefore, the ordering between Lines 229 and 230 had a minor impact on our model.

---

> > ### Author Response · Authors · 2024-11-25
> >
> > We sincerely thank the reviewer for the effort in reviewing our paper. We would greatly appreciate it if the reviewer let us know whether our response was helpful in addressing the reviewer's concerns. If there are additional concerns or questions, please let us know. If our responses were helpful in addressing concerns, we kindly ask the reviewer to consider raising the score.

---

### Meta-Review · Area_Chair_YGt7 · 2024-12-20

**Metareview:**

This paper introduces Simulation-free Entropic Unbalanced Optimal Transport (SF-EUOT), a scalable approach to solving the EUOT problem, a generalization of the Schrödinger bridges problem. By deriving its dynamical form, dual formulation, and optimality conditions, the method avoids costly simulations and enables one-step generation via the reciprocal property. SF-EUOT demonstrates scalability and good performances in generative modeling and image-to-image translation tasks compared to traditional Schrödinger bridge-based methods.

Reviewers expressed some concerns related to the general performances of the method wrt. other diffusion/distillation based methods, and the scientific novelty and added value of the proposed approach. Authors provided novel experiments, that are satisfying, and made more explicit the connections with the semi-relaxed unbalanced entropic OT problem. Overall, I believe that the paper might not be ready for publication at ICLR, given the amount of changes that the paper undergone during the rebuttal, and that would deserve a new review round. In this light, I will recommend a reject option, and I encourage the authors to resubmit a new version of their work, focusing eventually on the points raised by the reviewers, i.e. justifying and focusing on the impact of unbalancing the OT (outliers / generalization perspective), and comparisons with diffusion based approaches.

**Additional Comments On Reviewer Discussion:**

A lot of discussions happened between the reviewers and the authors, that yield new experiments and changes in the paper. Unfortunately, and as no reviewers champion this paper for acceptance, I believe the paper is not ready for publication yet.

---

### Decision · Program_Chairs · 2025-01-22

Reject